# An Adaptive Grasping Multi-Degree-of-Freedom Prosthetic Hand with a Rigid–Flexible Coupling Structure

**DOI:** 10.3390/s25196034

**Published:** 2025-10-01

**Authors:** Longhan Wu, Qingcong Wu

**Affiliations:** College of Mechanical and Electrical Engineering, Nanjing University of Aeronautics and Astronautics, Nanjing 210006, China; wulonghan@nuaa.edu.cn

**Keywords:** prosthetic hand, rigid–flexible coupling, grasping

## Abstract

**Highlights:**

**What are the main findings?**
A multi-degree-of-freedom prosthetic hand with a rigid–flexible coupling structure.Control algorithms for flexible object grasping.Mapping surface electromyographic (sEMG) signals to force using temporal convolutional network (TCN).

**What is the implication of the main finding?**
It can provide an additional adaptive joint for the finger without adding actuators.It is more suitable for grasping flexible objects than traditional algorithms.It provides a more accurate method to map sEMG signals of eight channels to force signals.

**Abstract:**

This study presents the design and evaluation of a dexterous prosthetic hand featuring five fingers, ten independently actuated joints, and four passively driven joints. The hand’s dexterity is enabled by a novel rigid–flexible coupled finger mechanism that incorporates a 1-active–1-passive joint configuration, which can enhance the dexterity of traditional rigid actuators while achieving a human-like workspace. Each finger is designed with a specific degree of rotational freedom to mimic natural opening and closing motions. This study also elaborates on the mapping of eight-channel electromyography to finger grasping force through improved TCN, as well as the control algorithm for grasping flexible objects. A functional prototype of the prosthetic hand was fabricated, and a series of experiments involving adaptive grasping and handheld manipulation tasks were conducted to validate the effectiveness of the proposed mechanical structure and control strategy. The results demonstrate that the hand can stably grasp flexible objects of various shapes and sizes. This work provides a practical solution for prosthetic hand design, offering promising potential for developing lightweight, dexterous, and highly anthropomorphic robotic hands suitable for real-world applications.

## 1. Introduction

The human hand possesses excellent dexterity and adaptability, enabling stable and efficient grasping of objects with different shapes, sizes, and stiffness [1]. Therefore, one of the key goals of humanoid robotic hands (especially prosthetic hands) is to mimic the adaptive grasping function of the human hand, thereby accomplishing diverse manipulation tasks in practical applications [2]. The realization of this function imposes higher requirements on the structural design and degree-of-freedom (DOF) configuration of prosthetic hand systems.

Full-drive solutions, such as the Shadow Dexterous Hand (24 DOFs), have achieved human-like manipulation. However, their weight and volume are far greater than those of a natural human hand. Additionally, their complex tendon-driven mechanisms require frequent maintenance, making them difficult to be deployed as practical prosthetic hands. In contrast, the prosthetic hand (19 DOFs) developed by Hao Yang realizes muscle-like contraction through closed-loop temperature control, thereby resolving the weight-flexibility trade-off inherent in traditional high-DOFs prosthetic hands [3]. However, its actuators remain concentrated in the forearm and fail to achieve modularity.

Regarding underactuated solutions, devices like the QB SoftHand (19 DOFs) simplify the mechanics by using a single motor to drive all fingers. Nevertheless, they fail in tasks that demand selective finger control, as their rigid transmissions cannot adapt to irregular object contours. Another design, SoftHand-A (2 DOFs), provides a solution for developing low-cost, minimally actuated hands but is only capable of grasping lightweight objects [4].

Rigid–flexible coupled prosthetic limbs typically achieve human-like grasping functions while maintaining a size similar to that of a natural human hand. Examples include the prosthetic hand (5 DOFs) developed by Xuan et al., whose finger joints adopt a rigid skeleton integrated with a flexible driving structure, enabling high adaptive grasping capability [5]. The OLYMPIC Hand (5 DOFs) addresses the design issues and limitations of current modular commercial and research-grade prosthetic hands, exhibiting excellent performance in grasping [6]. However, both designs suffer from a low number of DOFs and face challenges in achieving precise force control.

Currently, commercial and research-grade prosthetic hands rely heavily on sEMG signals. However, these systems exhibit high variability due to muscle fatigue, changes in skin-electrode contact, and environmental interference. Research by Eric et al. investigated the impact of feedback mechanisms on the control accuracy of myoelectric prostheses, noting that traditional myoelectric control systems suffer from error accumulation issues in virtual tasks [7]. Simon et al. mapped EMG signals to force signals using a three-dimension TCN, leveraging the TCN’s advantage of longer memory capacity compared to other recursive structures. This approach achieved a percentage root mean square error (%RMSE) of 29.2 ± 13.1% and a coefficient of determination (R^2^) of 69.4 ± 25.7% [8].

For researchers, grasping flexible objects with varying stiffness represents an important yet challenging task, as prosthetic hands struggle to autonomously determine the required degree of finger flexion to achieve the expected grasping force. Pimpalkar et al. noted in their research that the human hand can determine object stiffness within 15 milliseconds using vibrational signals at the moment of contact [9]. In contrast, existing prosthetic hands lack real-time stiffness sensing capabilities, resulting in delayed adjustment of grasping force relative to the object’s deformation speed. Traditional methods primarily rely on tactile force sensors or measuring hand displacement under applied force, but these approaches only become effective after full contact is established [10], increasing the risk of damaging fragile objects during delicate tasks.

To address this series of challenges, we have made the following innovations:This study drew inspiration from the structures of rigid dexterous hands and rope-driven dexterous hands [4], and combined the advantages of rigid dexterous hands and rope-driven dexterous hands to design a rigid–flexible coupling structure. This structure makes the joints of the fingers of the dexterous hand consistent with those of the human hand, and the degrees of freedom of the metacarpophalangeal (MCP) joints are directly driven by miniature electric cylinders. The proximal interphalangeal (PIP) joint is indirectly driven by a planar four-bar mechanism, and the Angle corresponds to that of the MCP joint. The distal interphalangeal joint is driven by a rigid–flexible coupling structure, which can adapt to the shape of the object without active control, thereby increasing the grasping area and enhancing the grasping stability.Meanwhile, a TCN-LIA-MPC algorithm is proposed. This algorithm enables the sEMG from the human forearm to be mapped into the human-expected grasping force via the Improved TCN algorithm. Subsequently, it obtains the expected displacement of the electric cylinders for grasping flexible objects using the linear iterative approximator (LIA) algorithm. Finally, it achieves effective tracking of the expected force through the Model Predictive Control (MPC) algorithm, thereby assisting humans in better grasping flexible objects.

## 2. Design of the Prosthetic Hand System

### 2.1. Human Hand Model Analysis and Reduction

The human hand exhibits exceptional grasping and manipulation capabilities, adapting seamlessly to objects of varying shapes and sizes while maintaining stable contact and force control. This high level of dexterity stems from the hand’s complex musculoskeletal structure and the spatial coordination among fingers. To emulate these features, numerous anatomical and kinematic models have been proposed, providing essential guidance for the design of prosthetic and anthropomorphic robotic hands [11].

Among widely accepted models (As shown in Figure 1a), the human hand is commonly described as having 23 DOFs: five in the thumb, four in each of the remaining fingers, and two additional DOFs located at the MCP joints of the ring and little fingers [12]. This model is extensively used in both biomechanical analysis and the development of multi-jointed robotic hands. Studies suggest that effective hand function relies heavily on the shared workspace among fingers—defined as the overlapping region of reachable fingertip trajectories [13].

Each finger’s flexion, extension, and adduction can be modeled as a serial kinematic chain with three DOFs, constrained by anatomical joint limits. As a result, the reachable workspace of a fingertip typically forms a crescent-shaped planar region [14]. To achieve effective grasping, it is critical to ensure substantial overlap between the workspaces of adjacent or opposing fingers. A larger shared workspace directly correlates with improved cooperative manipulation and adaptability to diverse objects [15].

Two key factors are essential for enhancing hand functionality: (1) maximizing the individual reachable area of each finger; and (2) enlarging the intersection of reachable regions between fingers to support coordinated grasping.

However, traditional rigid prosthetic hands face notable limitations in meeting these criteria. Most employ serial or parallel joint configurations with fixed kinematic structures, enabling only one-dimensional trajectory tracking rather than full-area coverage. Furthermore, their inherent rigidity restricts finger-to-finger compliance and coordinated adaptability, constraining both versatility and grasp stability [16,17].

To bridge the gap between traditional rigid prosthetic hands and the dexterity of the 23-degree-of-freedom human hand, two key design strategies are adopted. First, each finger is assigned four DOFs, which are essential for generating large individual fingertip workspaces. Second, a simplified palm mechanism with a single DOF is introduced to replace the four DOFs in the biological palm model. This palm joint enables thumb rotation and alignment, thereby forming a critical shared workspace among the fingers that is necessary for coordinated grasping.

Based on these simplifications, the proposed prosthetic hand model is shown in Figure 1b. It consists of 14 joints in total, including 10 actively actuated DOFs and 4 passively driven DOFs. Specifically, PIP joints are passively actuated and mechanically coupled with the MCP joints. This design reduces actuation complexity while preserving key functional synergies required for dexterous hand movements.

### 2.2. Design of Prosthetic Hand

The robotic prosthetic hand designed in this study consists of a palm and five fingers (as shown in Figure 2), with a total weight of 692.7 g, a height of 200.75 mm, a width of 96.5 mm, and a thickness of 39.5 mm. The core design concept lies in introducing motion coupling between the MCP and PIP joints, as well as between the PIP and distal interphalangeal (DIP) joints. This enables the four fingers—excluding the thumb—to achieve human-like flexible motion, thereby enhancing the ability to grasp and adapt to objects with complex curved surfaces, and supporting various manipulation modes such as pinching and gripping. All four fingers adopt the previously proposed rigid–flexible coupled design, with phalangeal segment lengths of 40.5 mm (proximal), 35 mm (middle), and 30 mm (distal). The thumb is designed as a six-bar linkage mechanism. The palm adopts a rigid base to provide structural support and accommodates various actuators to drive all five fingers.

The prosthetic hand is actuated by six linear cylinders and four AC servo motors. The selection criteria and specific functions of each actuator are described as follows:Two-phase four-wire AC motor CHF-GW12T-10BY (CHIHAI MOTOR, Huizhou, China): step Angle 18°, quantity 4. It achieves the lateral swing of the four fingers except the thumb through a worm gear reducer and also serves a reverse self-locking function. The self-locking mechanism can ensure that fingers do not rotate in the event of power failure or when grasping objects that are too heavy.Electric cylinder LA16-023D (INSPIRE-ROBOTS, Beijing, China): Stroke 16 mm, repeat positioning accuracy ±0.03 mm, maximum thrust 70 N, quantity 4. It drives the CAM through a slider mechanism, causing the four-bar mechanism at the proximal phalanx to move.Electric cylinder LAS16-023D (INSPIRE-ROBOTS): Stroke 16 mm, repeat positioning accuracy ±0.03 mm, maximum thrust 105 N, quantity 1. It achieves the relative rotation of the palm and the thumb through a lever mechanism.Electric cylinder LAS10-023D (INSPIRE-ROBOTS): Stroke 10 mm, repeat positioning accuracy ±0.02 mm, maximum thrust 70 N, quantity 1. It is responsible for driving the six-bar mechanism to achieve the flexion and extension of the thumb.

The selected electric cylinder has the advantages of small size and sufficient thrust, which is conducive to making our prosthetic structure more compact.

To achieve closed-loop control and environmental interaction, the prosthetic hand integrates multiple types of sensors:Position sensing: All electric cylinders are equipped with position feedback function, which can collect the current elongation of the electric cylinder in real time.Thin-film pressure sensor RP-C10-ST-LF2 (Xinbin Electronics, Shenzhen, China): Thin-film pressure sensors (with an outer diameter of 10 MM, short tail, low trigger range of 5 g–2 KG) are installed on each knuckle of the five fingers to detect the grasping force.

The system employs a laptop as the host controller for data processing and control algorithm execution, while utilizing an STM32F407VET6 microcontroller (168 MHz clock frequency) for force and position data acquisition.

The prototype of the prosthetic hand is shown in Figure 3. It can grasp a cylinder with a maximum diameter of approximately 11 cm.

### 2.3. Design of Rigid–Flexible Coupling Finger

During natural flexion of the human hand, a biomechanical coupling exists between the DIP and PIP joints. Moreover, when grasping objects, the DIP joint is capable of adaptively increasing its flexion angle based on the object’s shape, thereby enhancing contact stability. Inspired by this physiological phenomenon, this study proposes a novel rigid–flexible coupled robotic finger. The design aims to replicate the cooperative manipulation capabilities discussed earlier, while improving adaptability to objects with diverse geometries.

The two lines in the rigid–flexible coupling structure (as shown in Figure 4) can be regarded as the antagonistic tendon and the driving tendon, respectively.

### 2.4. Force Estimation Based on TCN

TCN effectively models temporal data through dilated convolutions and residual connections, demonstrating particular suitability for processing dynamic time-series signals such as sEMG signals [18]. In this study, the improved TCN algorithm will be used to process the sEMG signals from eight channels in real time, mapping the sEMG signals to the expected grasping force of the prosthetic hand.

The TCN architecture developed in this study is designed to predict force magnitude from 8-channel EMG signals, with its core advantages comprising the following:Long sequence modeling capability: By stacking dilated convolution layers, the TCN can cover several seconds of EMG signal history, effectively capturing the time-varying nonlinear relationship between force and EMG during dynamic grasping;Real-time performance: The causal convolution structure ensures that computation relies only on past data, meeting the low-latency requirements of prosthetic hand control;High parameter efficiency: Compared with Recurrent Neural Networks, TCN adopt shared convolution kernel weights, demonstrating stronger generalization in inter-subject training scenarios.

Dilated convolution serves as the fundamental operation in TCNs, which expands the receptive field by introducing “holes” (dilated intervals) between convolutional kernel elements. This operation is formally defined as(1)$$y[t]=\sum_{k}^{K-1}x[t-d\cdot k]\cdot w[k]$$
where y[t] is the value of the output sequence at time t, x[t] is the input sequence, w[k] is the weight of the convolution kernel (K is the kernel size), d is the dilation rate, which controls the receptive field range.

For L-layer dilated convolution (with an dilation rate of $d_{0},d_{1},\ldots ,d_{L-1}$), the total receptive field is(2)$$RF=1+\sum_{l=0}^{L-1}(K-1)\cdot d_{l}$$

In this paper, K=3, and the expansion rate increases exponentially by $d_{l}=2^{l}(l=0,1,2,3)$. The total receptive field of the 4-layer network is 31. It can capture the dependencies of 31 time steps.

Each temporal block contains two layers of dilated convolution, batch normalization, ReLU activation, and residual joint, which are mathematically expressed as(3)$$z_{1}=ReLU(BN(Conv1D(x,w_{1},d)))$$(4)$$z_{2}=ReLU(BN(Conv1D(z_{1},w_{2},d)))$$(5)$$y=ReLU(z_{2}+Downsample(x))$$

Among them, $Downsample\left(x\right)$ is a 1 × 1 convolution (when the number of input and output channels is different), ensuring that the dimensions of the residual connections match.

The model structure is shown in Figure 5, consisting of an input layer, four stacked temporal convolutional blocks, and an output layer:

Input layer: Receives filtered 8-channel sEMG signals with dimensions of  (N,1,8)(where N
represents the number of samples and 1 represents the number of input channels).Temporal convolution block: Block 1: d = 1, output channel = 16, receptive field = 3; Part 2: d = 2, Output channel = 16, receptive field = 7; Section 3: d = 4, Output channel = 16, receptive field = 15; Section 4: d = 8, output channel = 16, receptive field = 31.Output layer: The feature dimensions are transformed into (N, 128) (16 × 8) through flattening operations, and the force prediction value F
is output through the fully connected layer.

The obtained force F is the combined force exerted by the proximal phalanges, middle phalanges and distal phalanges, that is, the grasping force acting on the object. The force analysis is shown in Figure 6.(6)$$F=F_{d}+F_{p}+F_{m}$$

### 2.5. Dynamics Analysis of Rigid–Flexible Coupling Finger

This section aims to elaborate on the method for obtaining position information of each joint based on the elongation of the electric cylinder push rod and realizing the grasp force estimation of a single mechanical finger.

The movement of the proximal phalanx is accomplished by the electric cylinder driving the slider–rocker mechanism to drive the cam and thus push the phalanx. Therefore, there is a deterministic mapping relationship between its rotation angle $\theta_{m}(t)$ and the absolute position of the push rod. We use inertial measurement unit (IMU) as the tool for verifying the model accuracy. The joint position data corresponding to different push rod elongations are collected by an IMU, and the functional relationship between the two obtained through fitting is as follows (Polynomial fitting order: 8, MSE: 0.065760, R^2^: 0.999775):(7)$$\theta_{m}\left(t\right)=\left\{\begin{array}{c} 6.1036\times 10^{-5}x^{8}-0.0026x^{7}+{0.0457x}^{6}-0.4019x^{5}+1.8795x^{4}\\ \;-4.3129x^{3}+3.7159x^{2}+0.8483x^{1}-1.8907,\;x\geq 2.87\\ 0,\;x<2.87 \end{array}\right.$$

Here, x represents the absolute displacement of the electric cylinder.

The relative rotation angle of the middle phalanx $\theta_{p}\left(t\right)$ is consistent with the rotation angle corresponding to $l_{2}$ in Figure 7. Since the MCP joint and PIP joint are transmitted through a four-bar linkage, their angular relationship satisfies:(8)$$l_{4}(t)=\sqrt{l_{0}^{2}+l_{1}^{2}-2l_{3}l_{4}cos(\pi -\theta_{m}(t))}$$(9)$$\varphi_{1}(t)=arcsin(\frac{l_{1}sin(\pi +\theta_{m}(t))}{l_{4}(t)})$$(10)$$\varphi_{2}\left(t\right)=arccos(\frac{l_{3}^{2}+l_{4}{\left(t\right)}^{2}-l_{2}^{2}}{2l_{3}l_{4}(t)})$$(11)$$\theta_{p}\left(t\right)=arcsin(\frac{l_{1}sin\theta_{m}\left(t\right)-l_{3}sin(\varphi_{2}\left(t\right)-\varphi 1(t))}{l_{2}})$$

Among them, $l_{0}=6.73\;mm,l_{1}=36.42\;mm,l_{2}=9\;mm,l_{3}=39\;mm$.

Based on the previously established kinematic mapping between the absolute position of the push rod and PIP joint angle, Figure 8a,b reveals the quantitative relationship between the push rod displacement and the PIP joint angle. Due to factors such as gaps during manufacturing and installation, as the absolute value displacement of the electric cylinder increases, the rotational error of the mapped PIP joint and the error of the theoretical model increase accordingly. Therefore, it is necessary to multiply the theoretical value by a coefficient *k* = 1.15 to reduce the absolute value error during control [19]. The modified equation is(12)$$\theta_{p}\left(t\right)=k\;arcsin(\frac{l_{1}sin\theta_{m}\left(t\right)-l_{3}sin(\varphi_{2}\left(t\right)-\varphi 1(t))}{l_{2}})$$

The rotation angle of the distal phalanx, $\theta_{d}\left(t\right)$, is directly related to the deformation of the spring in the rigid–flexible coupled structure: the spring is pulled by the middle phalanx, and its deformation increases with the increase in the PIP joint angle.

According to the geometric constraints of the finger coupling mechanism, an analytical relationship between the DIP joint angle ${\;\theta }_{d}\left(t\right)$ and the PIP joint angle under no-load conditions can be derived. By substituting the previously established function relating the absolute position of the push rod and the PIP joint angle, a mapping curve between the absolute position of the push rod and the DIP joint angle $\theta_{d0}\left(t\right)$ under no-load conditions can be obtained, as shown in Figure 8c.

The simplified diagram of the load at the end of the mechanical finger is shown in Figure 9. Under loading conditions, the DIP joint angle $\theta_{d}\left(t\right)\;$ must be solved numerically using an iterative method:(13)$$l_{e}=\mathrm{acos}\frac{2.2143-\theta_{d}(t)}{2}+r$$(14)$$\theta_{d}\left(t\right)=arccos\frac{{\left(\frac{F_{d}l_{d}}{kl_{e}}+a\sqrt{2-3\cos\theta_{d0}\left(t\right)}\right)}^{2}-2a^{2}}{2a^{2}}$$

### 2.6. Actuation and Control of Prosthetic Hand

Due to the significant diversity in the mechanical models of flexible objects (such as differences in material properties and geometric configurations), it is difficult to establish a universally applicable precise analytical model for each type of object; furthermore, even if modeling for known objects is completed, the robot may still produce unexpected motions when confronted with new unmodeled flexible targets [20].

To address this issue, we innovatively employed a position predictor in the control of rigid–flexible coupled fingers. The core of the position predictor lies in the online linear iterative approximation algorithm.

Let the initial contact point (without external force) be (0, $X_{e0}$), and the subsequent two contact state points be ($F_{e1}$, $X_{e1}$) and ($F_{e2}$, $X_{e2}$), respectively. Then the external force applied to the object and the corresponding deformation can be expressed as ($F_{e1}$, $X_{e1}-X_{e0}$) and ($F_{e2}$, $X_{e2}-X_{e0}$). Among them, $F_{e1}\;$ and $F_{e2\;}$ are obtained by vector superposition of the contact forces of the proximal phalanx ($F_{m}$), middle phalanx ($F_{p}$), and distal phalanx ($F_{d}$) at the corresponding moments, respectively. The direction angles of each force can be derived through the mapping relationship between the absolute position of the electric cylinder and the joint angles; $X_{e0}$, $X_{e1}\;$ and $X_{e2}\;$ correspond to the elongation of the micro electric cylinder, respectively.

Based on the above definitions, the approximate linear mathematical models for two different objects can be established as follows:(15)$$X_{d1}=\frac{X_{e2}-X_{e0}}{F_{e2}}F_{r}+X_{e0}$$(16)$$X_{d2}=\frac{X_{e2}-X_{e1}}{F_{e2}-F_{e1}}(F_{r}-F_{e2})+X_{e2}$$

Here, $X_{d1}$ and $X_{d2}$ represent two different types of predicted expected displacements, and $F_{r}$ is the expected force. The weighted predicted displacement $X_{d}$ can be calculated through $X_{d1}$ and $X_{d2}$ by the following formula:(17)$$X_{d}=X_{d1}\cdot k_{1}+X_{d2}\cdot (1-k_{1})$$

Among them, $k_{1}$ is the weight factor. When the next moment arrives, ($F_{e1}$,$X_{e1}$) will be replaced by ($F_{e2}$,$X_{e2}$), and ($F_{e2}$,$X_{e2}$) will be replaced by ($F_{c}$,$X_{c}$). Here, $F_{c}$ and $X_{c}$ are the latest force generated by the target and the latest position of the electric cylinder. After that, the new predicted displacement of the electric cylinder will be calculated for the next control cycle.

To meet the high-precision tracking requirements of the expected displacement trajectory $x_{d}(t)$ for prosthetic fingers, this study adopts Model Predictive Control (MPC). Its core advantages lie in the following aspects:Multi-step optimization capability: Through rolling horizon optimization, MPC can compensate for system inertia and external disturbances in advance, avoiding the lag issues inherent in traditional PID control;Explicit constraint handling: Physical constraints such as joint limits are directly embedded into the optimization problem, ensuring control safety;Model dependence: By utilizing the dynamic model of the controlled system to predict future states, MPC significantly enhances robustness against nonlinear friction and flexible deformation during the grasping process.

The discrete-time dynamic model of the electric cylinder is constructed based on the first-order inertial characteristics, and the state update formula is(18)$$x\left(k+1\right)=a\cdot x\left(k\right)+\left(1-a\right)\cdot u(k)$$

Among them: $x\left(k\right)$ is the actual position of the electric cylinder at time k, u(k) is the control instruction at time k, a is the tracking coefficient. The state constraint is $x\in [x_{min},x_{max}]$.

MPC predicts future states through rolling optimization. The prediction equation is extended based on Equation (17), and the predicted states for the next $N_{p}$ steps are(19)$$x\left(k+i|k\right)=a\cdot x\left(k+i-1|k\right)+\left(1-a\right)\cdot u\left(k+i-1|k\right)$$

Among them: $x\left(k+i|k\right)$ is the predicted position at time k+i based on the state at time k, $u\left(k+i-1|k\right)$ is the control instruction at time k+i−1 optimized at time k, When $i>N_{c}$, the control instruction maintains the last optimized value: $u\left(k+i-1|k\right)=u\left(k+N_{c}-1|k\right)$

The cost function contains four weighted penalty terms to balance tracking accuracy and control smoothness:(20)$$J=J_{1}+J_{2}+J_{3}+J_{4}$$

Tracking error penalty $J_{1}$: Minimize the deviation between the predicted trajectory and the current target position.(21)$$J_{1}=\sum_{i=1}^{N_{P}}Q\cdot (x\left(k+i|k\right)-x_{d}(k))$$

Here, Q is the tracking weight and $x_{d}(k)$ is the target position at time k.

Control quantity penalty $J_{2}$: Excessive suppression control instructions.(22)$$J_{2}=\sum_{i=1}^{N_{c}-1}R\cdot {\left(u\left(k+i|k\right)\right)}^{2}$$

Here, R represents the control weight.

Control rate of change penalty $J_{3}$: Reduce instruction mutations and ensure smoothness.(23)$$J_{3}=\sum_{i=0}^{N_{c}-2}R_{d}\cdot {\left(u\left(k+i+1|k\right)-u\left(k+i|k\right)\right)}^{2}$$

Here, $R_{d}$ represents the weight of the rate of change.

Overshoot penalty $J_{4}$: Suppresses forward overshoot beyond the target position.(24)$$J_{4}=w_{0}\sum_{i=1}^{N_{P}}\max\left(x\left(k+i|k\right)-x_{d}(k)\right)$$

Here, $w_{0}$ represents the overshoot penalty coefficient.

The control instructions must meet the physical stroke limit:(25)$$u\left(k+i|k\right)\in [x_{min},x_{max}]$$

### 2.7. Control Block Diagram

Overall, in this study, the force signal mapped to the sEMG signals was utilized as the expected force. LIA was used to track the expected force to obtain an expected displacement. Then, the expected displacement was predicted and controlled through MPC. The control block diagram is shown in Figure 10.

## 3. Experiments and Results

### 3.1. Calibration of Electromechanical Signals

To establish a quantitative mapping between sEMG amplitude and finger grasping force, the following experimental protocol was designed in this study:

A cylindrical rubber column was selected as the grasping target, with a diameter of 75 mm and a height of 90 mm. Three thin-film pressure sensors were affixed circumferentially to the surface of the rubber column, corresponding to the palmar contact areas of the thumb, index finger, and middle finger (as shown in Figure 11), respectively, to measure the normal contact force applied by each finger in real time. All sensors were statically calibrated using a standard force source to obtain force–voltage conversion coefficients, and were connected to a multi-channel data acquisition system.

During the experiment, participants were seated comfortably: the tested arm rested naturally on the table, with the elbow flexed at approximately 90°, and the wrist in a neutral position. A Myo armband was worn on the forearm to collect 8-channel sEMG signals. Low-pass filtering is adopted to preprocess the original 8-channel sEMG signals to eliminate noise. The pressure sensor signals and sEMG signals were time-aligned using a host computer, and the sampling rate was uniformly set to 100 Hz.

To ensure the normal operation of the model, the following two-step verification process needs to be implemented:Offline model optimization. The TCN model is trained on a large number of sEMG-force datasets, during which hyperparameters are continuously modified to minimize %RMSE and maximize R^2^.Online performance verification. After the training, we selected five subjects (three men and two women, aged 25 to 35, weighing 55 to 75 kg, with no history of forearm muscle injury or neuromuscular disease) for testing to confirm the real-time effectiveness. The key is to check whether the predicted force trend is consistent with the actual force trend.

The experimental results obtained are shown in Figure 12. The measured %RMSE is 14.66%, and the Validation R^2^ is 0.7003. Compared with other researchers (e.g., Simon et al. [8], who achieved %RMSE is 29.2% and R^2^ is 64.9 in their study), the improved TCN algorithm exhibits better performance in mapping eight-channel sEMG signals to expected forces.

### 3.2. Flexible Grasping Experiment

After the offline training of the EMG-to-force calibration model was completed, the subject wore the same Myo electromyographic acquisition device to collect real-time sEMG signals from target forearm muscles, at a sampling frequency of 100 Hz. The experimental scene of flexible grasping is shown in Figure 13. A pre-trained TNC model was used to map temporal sEMG features to continuous finger grasping force predictions in real time. These predicted forces served as real-time input commands for the control system.

Subsequently, the optimal control signals derived from MPC are used to drive the electric cylinder in real time, achieving precise displacement trajectory tracking. This ultimately enables constant-force grasping (with a target force of 2 ± 0.3 N) or variable-force grasping of flexible objects.

Throughout the entire experiment, the following data were recorded synchronously for effect verification: (1) the absolute displacement of the electric cylinder, (2) the grasping force prediction output by the TNC model, and (3) the actual contact force measured by the thin-film pressure sensor. The time-aligned results are shown in Figure 14a,b.

### 3.3. Comparative Experiment

Compared with impedance control

To evaluate the effectiveness of a linear controller in flexible object grasping tasks, after completing the original myoelectric-driven closed-loop grasping experiment, the configuration of the sEMG sensors and the target object was kept unchanged, while the linear controller in the original control strategy was replaced with an impedance controller. The same subject repeated the grasping process, and the root mean square error, maximum error, and average error under the two control algorithms were compared to evaluate the specific impact of control compliance on grasping stability.

B.Compared with PID control

To assess the performance of the MPC algorithm in flexible grasping, based on the original grasping experiment, the sEMG sensor configuration and target object were kept unchanged, and the MPC controller was replaced with a conventional PID controller. The same subject repeated the grasping process, and the root mean square error, maximum error, and average error under the two control algorithms were compared to evaluate the specific impact of control compliance on grasping stability.

After the above comparative experiments, the experimental results are shown in Table 1.

## 4. Discussion

Experimental results (Table 1) demonstrate that, when grasping the same deformable object, the force tracking accuracy of the LIA-MPC hybrid control strategy is significantly superior to that of conventional admittance control and the LIA-PID scheme, regardless of constant-force or dynamic variable-force grasping. Taking sponge grasping as an example, under constant expected force, RMSE of LIA-MPC is reduced by approximately 56.9% compared to admittance control and by approximately 17.5% compared to the LIA-PID scheme. Under a constant expected force, the standard deviation of the LIA-MPC is reduced by approximately 59.35% compared to the admittance control and by approximately 33.05% compared to the LIA-PID scheme. These results indicate that both LIA and MPC can achieve better force tracking accuracy when grasping deformable objects, thereby validating the effectiveness of the proposed algorithm in scenarios involving deformable object manipulation.

Notably, control performance exhibits significant differences depending on the stiffness of the deformable object. For low-stiffness objects (e.g., sponge block, cotton doll), the force tracking RMSE reaches 0.4209 and 0.4500 and the standard deviation reaches 0.3147 and 0.2494, respectively, whereas for high-stiffness objects (e.g., silica gel column, empty plastic water bottle), the RMSE increases to 1.0317 and 1.3085 and the standard deviation increases to 0.6727 and 0.7569, respectively. This discrepancy may arise from the following factors: (1) low-stiffness objects provide greater contact deformation space, which facilitates error compensation within the control system; (2) the contact nonlinearity (e.g., bouncing effects) associated with high-stiffness objects exacerbates force control oscillations.

Despite the significant improvement in control accuracy, persistent fluctuations in the force signal were observed during experiments. We attribute this phenomenon to two primary mechanisms:(1)Lack of multi-finger coordination: During grasping, independent control of the five fingers lacks a force coordination algorithm. When the deformable object shifts within the hand, asynchronous changes in contact conditions across the fingertip film pressure sensors lead to conflicts in force distribution.(2)The sensitivity of the sensors used is relatively low; consequently, the minimum measurable contact force may already cause significant displacement of the object. Therefore, the actually measured force may be inaccurate, and errors may further be introduced due to the low sensor sensitivity [21].(3)In traditional force-position control, unknown or varying environmental stiffness leads to steady-state force errors [22].

In future work, we will incorporate methods proposed by other researchers to address this issue. For instance, we will adopt their proposed unscented Bayesian optimization method to systematically tackle the uncertainties introduced by sensors; meanwhile, we will utilize a grey prediction model to predict environmental stiffness in real time, thereby reducing force tracking errors.

In terms of structure, Table 2 lists other research-based or commercial prosthetic hands or dexterous hands. Compared with some other rigid–flexible coupled prosthetic hands (e.g., Xuan’s prosthetic hand, OLYMPIC Hand), the prosthetic hand developed in this study exhibits more DOFs, theoretically enabling more types of movements, which will be explored in subsequent studies. In other fields, when compared with humanoid dexterous hands, although our prosthetic hand is inferior to those of some dexterous hands (e.g., Shadow Hand), it has advantages in terms of size and weight, and does not require driving a large number of actuators. This facilitates the modular design of the prosthetic hand and the development of control algorithms, making it more suitable for use as a human-wearable prosthetic limb. Meanwhile, the distal knuckles of the prosthetic hand in this study possess adaptive capabilities, overcoming the limitation that traditional rigid dexterous hands (e.g., Schunk SVH) are unable to adaptively grasp objects.

## 5. Conclusions

The grasping of flexible objects presents significant challenges for prosthetic hands due to the complex structures and heterogeneous material properties of such objects, which render explicit physical modeling and real-time contact force control inherently difficult. Moreover, excessive contact stress during grasping may cause permanent damage to delicate deformable objects.

To address these limitations, this study proposes a novel prosthetic hand system specifically designed for flexible object manipulation. The design incorporates 10 active degrees of freedom and 4 passive DOFs, integrating rigid–flexible coupled fingers with adaptive distal structures. This architecture enhances contact area during grasping without additional actuators, thereby improving stability while minimizing mechanical complexity. A functional prototype was developed and experimentally validated, demonstrating its capability to handle irregular objects with human-like adaptability.

For control implementation, we developed an improved TCN framework to map sEMG signals to expected grasping forces. This was combined with a linear predictor and MPC strategy to achieve accurate force tracking. Comparative experiments against conventional impedance and PID controllers confirmed the superior performance of the proposed approach. Extensive grasping trials on various flexible objects demonstrated a significant improvement in force control precisions.

This work contributes to soft robotics and prosthetic design by (1) introducing a mechanically adaptive hand structure that reduces modeling dependency for unknown objects; (2) establishing a deep learning-based force calibration framework that enhances sEMG-force mapping robustness; and (3) validating a hybrid control strategy that ensures stable interaction with deformable materials. Future research will focus on further improving the mapping accuracy of TCN and the control precision of control algorithms, as well as demonstrating the superiority of rigid–flexible coupled structures over rigid structures through data analysis.

## Figures and Tables

**Figure 1 sensors-25-06034-f001:**
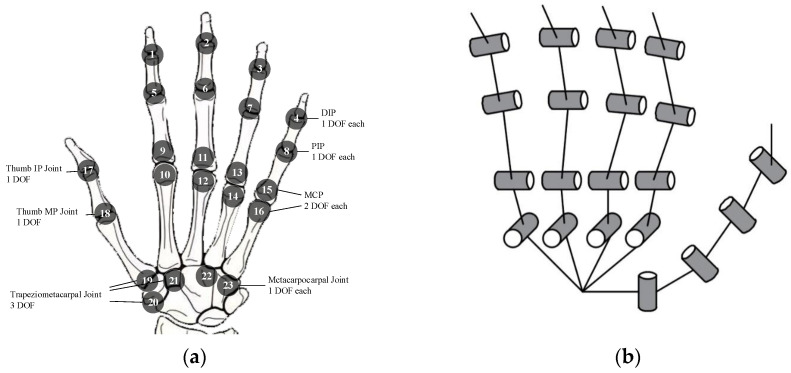
Simplification of the Human Hand Model (**a**) 23-DOFs model of the human hand. (**b**) The simplified model of 10 active degrees of freedom and 4 passive degrees of freedom for the proposed rigid–flexible coupled dexterous hand. Each finger’s MCP joint is driven by a motor, the PIP joint is a passive joint, and the DIP joint is a passive degree of freedom.

**Figure 2 sensors-25-06034-f002:**
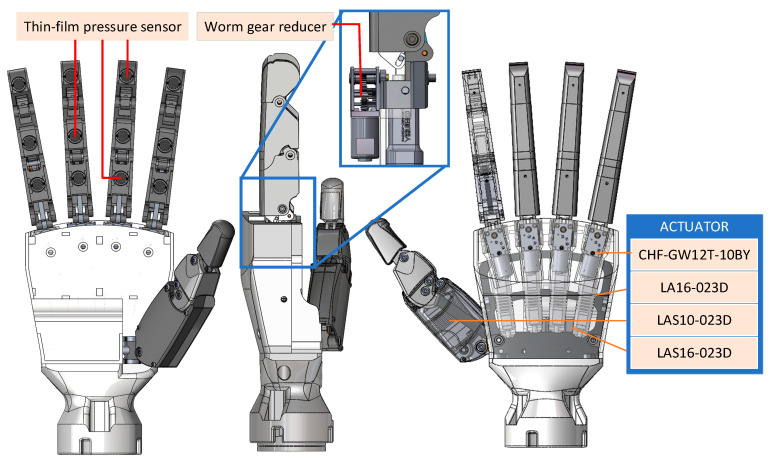
Schematic diagram of the overall structure of the prosthetic hand. The prosthetic hand has five fingers. Among them, the thumb is driven by the electric cylinder LAS10-023D, and the relative rotation of the palm and thumb is driven by LAS16-023D. The remaining four fingers are driven by the electric cylinder LA16-023D and the motor CHF-GW12T-10BY. Thin-film pressure sensors are evenly distributed on the three knuckles of the five fingers.

**Figure 3 sensors-25-06034-f003:**
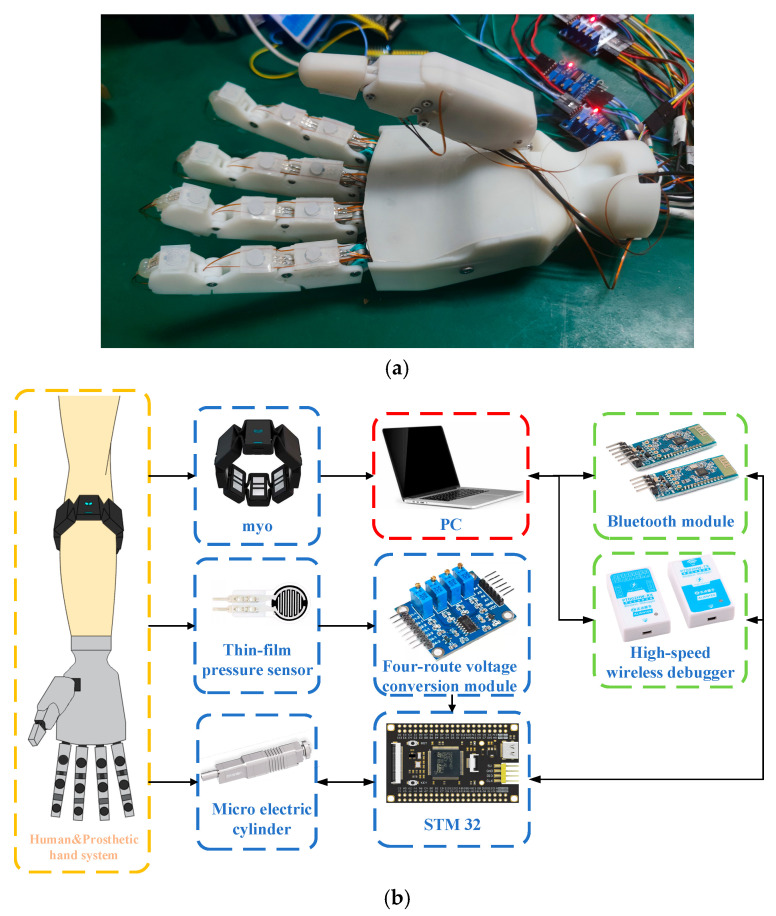
(**a**) The prototype of prosthetic hand. (**b**) Configuration of the prosthetic hand control platform. The eight-channel sEMG signals collected by myo is directly sent to the PC, which is mapped to the expected force by the improved TCN trained in the PC. Meanwhile, the force signal collected by the thin-film pressure sensor and the current displacement of the electric cylinder are collected by STM32 and sent to the PC through the wireless module. After processing the actual force, actual displacement and expected force, the PC obtains an expected displacement. It is sent to the STM32 through the wireless module, and the STM32 drives the electric cylinder to move.

**Figure 4 sensors-25-06034-f004:**
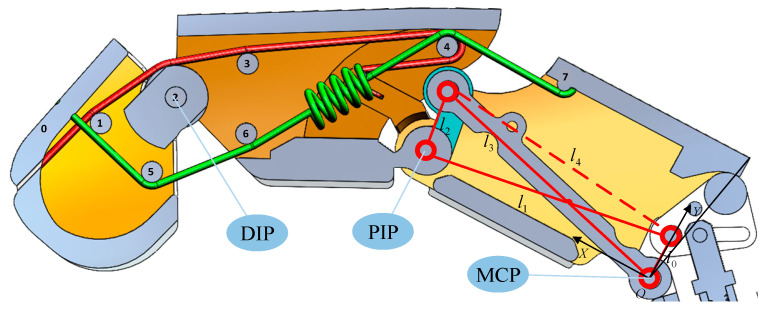
Schematic diagram of the rigid–flexible coupling finger structure. The antagonistic tendon is located above, and it starts from Shell 0, successively bypasses Optical Axis 1, Optical Axis 2, Optical Axis 3, and Optical Axis 4, and reaches Shell 7. It is responsible for resetting the fingers. Below is the driving tendon, which starts from Shell 0, passes around Optical Axis 1, Optical Axis 5, Optical Axis 6, and Optical Axis 4 in sequence, and reaches Shell 7. A spring is connected between Optical Axis 6 and Optical Axis 4. It is responsible for achieving the inward rotation of the end joint.

**Figure 5 sensors-25-06034-f005:**
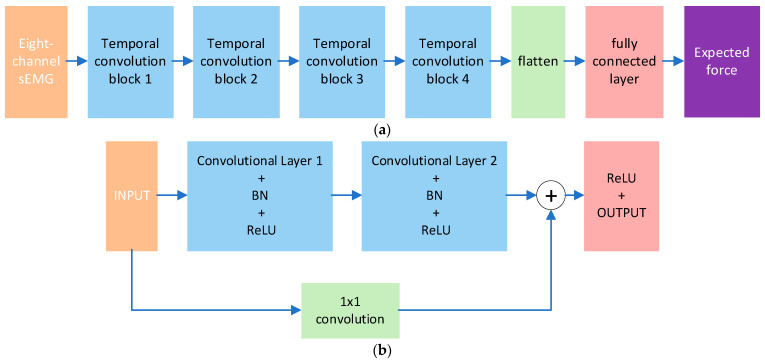
(**a**) The overall architecture of improved TCN. Input eight channels of sEMG signals and output the expected force. (**b**) Temporal convolutional block structure.

**Figure 6 sensors-25-06034-f006:**
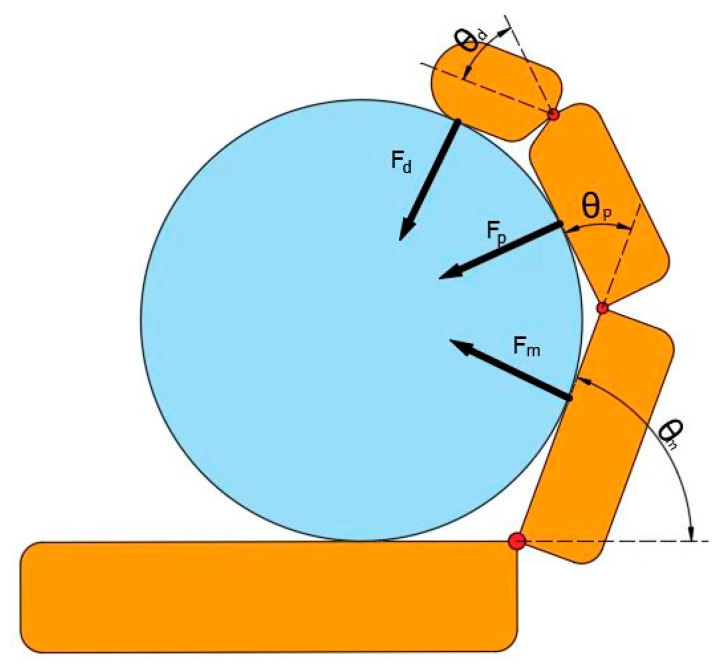
Simplified diagram of mechanical finger grasping force. Among them, the blue part represents the problem of being grasped, and the orange part represents the palm and fingers of the prosthetic hand.

**Figure 7 sensors-25-06034-f007:**
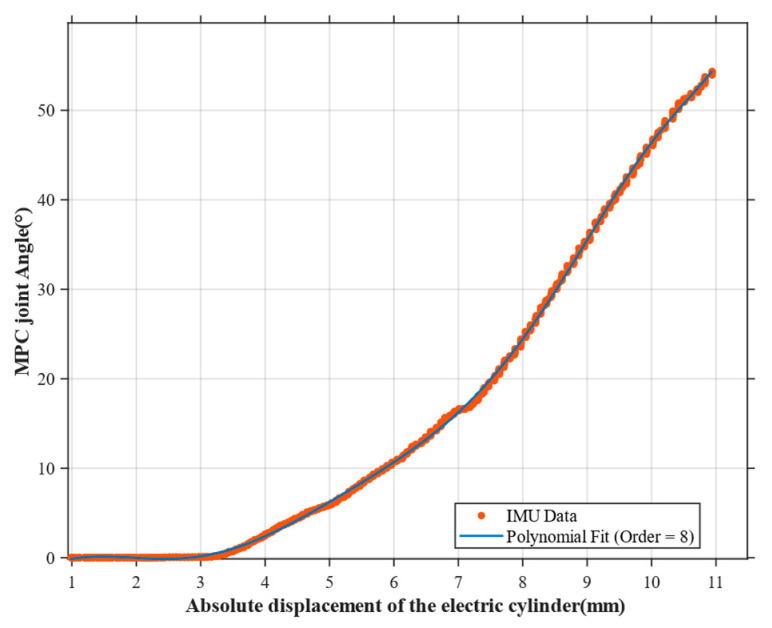
The mapping relationship between the absolute value position of the electric cylinder and the MPC joint.

**Figure 8 sensors-25-06034-f008:**
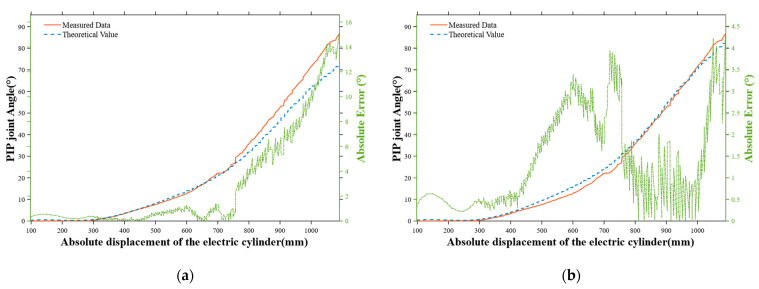
(**a**) Comparison between theoretical values and measured data of the PIP joint, the case without the coefficient k. (**b**) Comparison between theoretical values and measured data of the PIP joint, the case with the coefficient k. (**c**) Comparison between theoretical values and measured data of the DIP joint.

**Figure 9 sensors-25-06034-f009:**
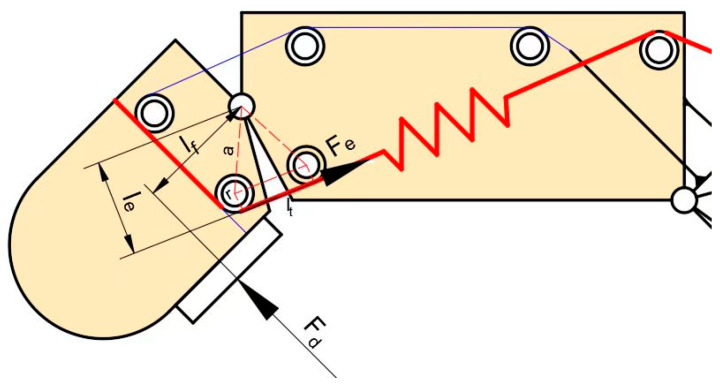
A simple diagram of the load at the end of a mechanical finger.

**Figure 10 sensors-25-06034-f010:**
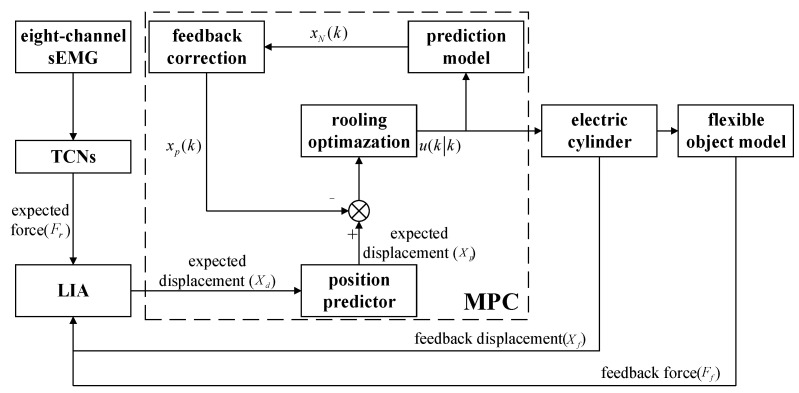
The control block diagram.

**Figure 11 sensors-25-06034-f011:**
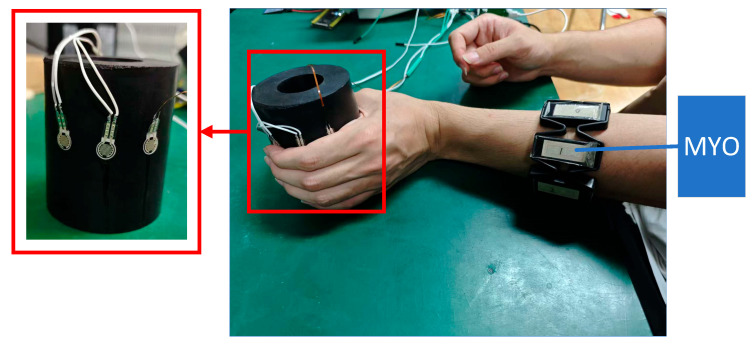
TCN experimental scene.

**Figure 12 sensors-25-06034-f012:**
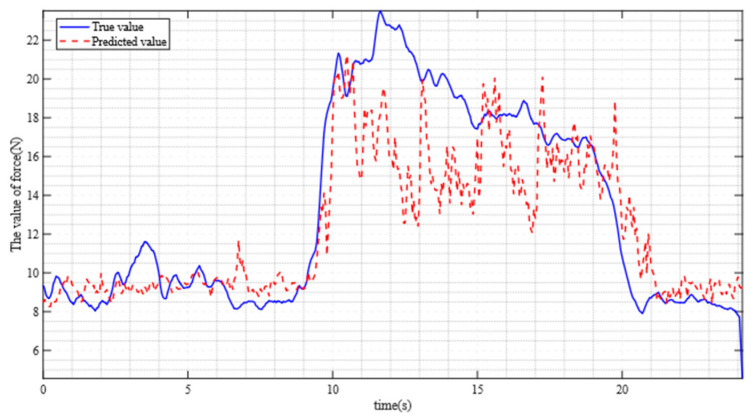
Comparison between predictive force and actual force.

**Figure 13 sensors-25-06034-f013:**
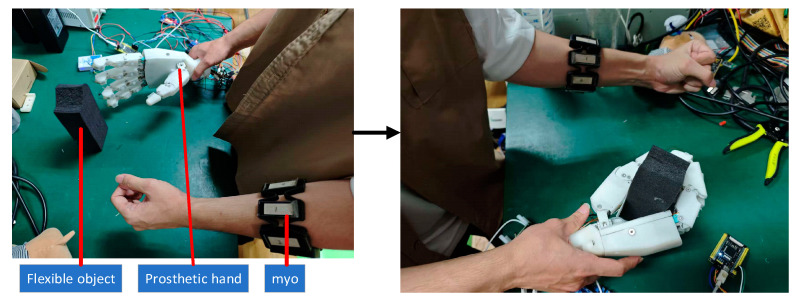
Flexible grasping experimental scene.

**Figure 14 sensors-25-06034-f014:**
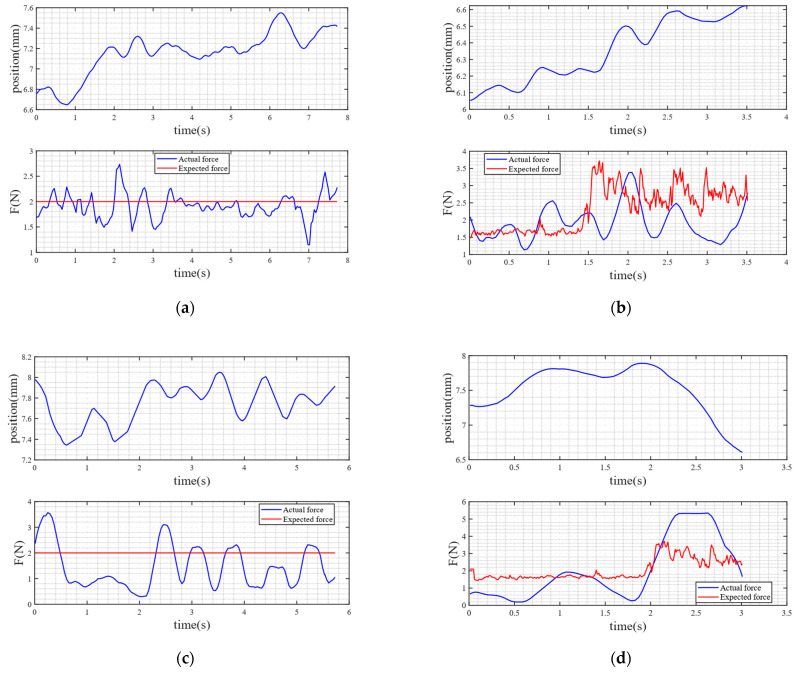
The absolute displacement of the electric cylinder and the tracking of the actual force to the expected force in various modes. (**a**) LIA-MPC under constant force; (**b**) LIA-MPC under variable force; (**c**) AC under constant force; (**d**) AC under variable force; (**e**) LIA-PIP under constant force; (**f**) LIA-PIP under variable force.

**Table 1 sensors-25-06034-t001:** Comparative experiment.

Grasping Object	Mode	Expectation Force	RMSE	Max Error	Mean Error	Standard Deviation
Sponge block	LIA + MPC	2 N	0.4209	1.0489	0.2150	0.3147
		changing	0.8068	2.2061	0.4337	0.6955
	AC	2 N	0.9767	1.6973	0.5359	0.7742
		changing	1.7685	3.6155	−0.9912	0.6063
	LIA + PID	2 N	0.5103	1.0300	0.2495	0.4701
		changing	1.7726	4.3854	0.2797	1.5344
Cotton doll	LIA + MPC	2 N	0.4500	1.3273	0.1816	0.2494
		changing	1.3537	3.9274	0.5565	0.7517
	AC	2 N	0.8077	2.4817	0.6917	0.5512
		changing	1.5234	2.5414	1.0891	0.7839
	LIA + PID	2 N	0.9131	1.5438	0.3532	0.8556
		changing	1.6440	4.5594	0.7784	1.2610
Silica gel column	LIA + MPC	3 N	1.0317	2.0128	−0.2345	0.6727
		changing	1.2478	2.2787	−0.6884	1.0170
	AC	3 N	1.9314	2.7228	1.1697	0.9033
		changing	2.6348	4.9876	0.7574	0.7078
	LIA + PID	3 N	1.0837	2.2783	0.6289	1.0410
		changing	2.0797	4.3806	0.3062	1.4421
Empty plastic water bottle	LIA + MPC	3 N	1.3085	3.8693	−0.0790	0.7569
		changing	1.1668	3.3395	0.6302	1.0801
	AC	3 N	1.7759	2.7886	1.0125	0.5789
		changing	1.1655	3.1328	0.4134	1.1949
	LIA + PID	3 N	1.2319	2.4915	0.6946	0.9461
		changing	1.8022	3.8478	0.9297	1.3424

**Table 2 sensors-25-06034-t002:** Comparison of high-DOF bionic hands.

Name	Type	Author/Company	Number of DOFs	Number of Actuators
Our prosthetic hand	Rigid–flexible coupled prosthetic hand	our study	14	10
Xuan’s prosthetic hand	Rigid–flexible coupled prosthetic hand	Xuan, S et al. [5]	5	5
OLYMPIC Hand	Rigid–flexible coupled prosthetic hand	Liow, L et al. [6]	5	5
Schunk SVH	Rigid dexterous hand	SCHUNK (Stuttgart, Germany)	9	9
Shadow hand	Tendons drive dexterous hands	Shadow Robot (London, UK)	24	20
SoftHand-A hand	Underactuated dexterous hand	Li, H et al. [4]	5	2
QB SoftHand	Underactuated dexterous hand	qb robotics (Cascina, Italy)	19	2

## Data Availability

The datasets used and analysed during the current study are available from the corresponding author on reasonable request.

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
