# Peer review of "An Adaptive Grasping Multi-Degree-of-Freedom Prosthetic Hand with a Rigid–Flexible Coupling Structure"

_sensors, 2025, doi:10.3390/s25196034_

Round 1
Reviewer 1 Report
Comments and Suggestions for Authors
This article proposes an adaptive multi degree of freedom grasping dexterous hand, completes the overall structural design and prototype integration, and proposes a compliant control method called TCN-LIA-MPC. The theory, methods, experiments, and related discussions are rich and have good engineering application value, but there are some minor issues that need to be corrected:
- Suggest labeling Figure 4 with necessary key components, such as indicating the location of MCP.
- There is a duplicate "Tem. Con. Block 2" in Figure 5 (a). It is recommended that the author double check for errors.
- Please correct some minor errors, such as "FigureX" on line 228, two xd (t) on line 304,% RMSE on line 368, and "Fig. X and Table X" on line 413.
- Suggest supplementing the configuration of the dexterous hand control platform. How is the coordination of motor control under so many degrees of freedom?
- Is the TCN algorithm used for real-time processing of EMG signals from 8 channels? Suggest adding necessary explanations in the text on how the algorithm is specifically implemented.
- Is the signal from the IMU sensor used for closed-loop control of the movement of the Reiki hand when " The joint position data corresponding to different push 239 rod elongations are collected by an inertial measurement unit (IMU)," is mentioned in the article?
- According to the running results shown in Fig.8, after opening parameter k, the absolute error significantly increased. Please explain the detailed reasons in conjunction with the proposed control algorithm.
Author Response
Dear reviewer #1,
Thank you very much for giving us an opportunity to revise the manuscript sensors-3858169. According to your valuable comments, the manuscript has been carefully revised, and the major revisions were marked in red. We sincerely thank you for your careful read of the paper and helpful suggestions in improving the quality of our submission. The following are the detailed replies to your comments and the changes made in the revision:
Comment 1: Suggest labeling Figure 4 with necessary key components, such as indicating the location of MCP.
Response 1: We fully understand the reviewer’s concern and thank for this comment. We added clear labels to the key components in Figure 4. Including clearly marking the positions of metacarpophalangeal joint (MCP), Proximal interphalangeal joint (PIP), and Distal Interphalangeal joint (DIP). Avoid blurring the structural details of the prosthetic hand to ensure readability.
Comment 2: There is a duplicate "Tem. Con. Block 2" in Figure 5 (a). It is recommended that the author double check for errors.
Response 2: We fully understand the reviewer’s concern and thank for this comment. The duplicate label "Tem.Con.Block 2" has been corrected to" Tem.Con.Block 3". In addition, we also checked all the other elements in Figure 5 (for example, the text in the module, the arrow points) to ensure that there are no other similar label errors.
Comment 3: Please correct some minor errors, such as "FigureX" on line 228, two xd (t) on line 304, % RMSE on line 368, and "Fig. X and Table X" on line 413.
Response 3:Thank you for pointing out these mistakes. We have verified and corrected each issue as follows:
(1) The incomplete "FigureX" in line 228 has been updated to "Figure 6" in line 271.
(2) The redundant "xd (t)" in line 348 has been deleted.
(3) After verification, “% RMSE” (Percentage Root Mean Square Error) is indeed the standard and intentional notation rather than an error. As a widely used metric in regression tasks (e.g., the desired force prediction in our study), % RMSE normalizes the root mean square error into a percentage of the target variable’s range (or mean value). This standardization enables intuitive comparison of model performance across different datasets or with other studies in the dexterous robotic hand field—consistent with common practices in related literature.
(4) The incorrect "Fig. X and Table X" in line 413 has been updated to "Table 1" (in line 467).
Comment 4: Suggest supplementing the configuration of the dexterous hand control platform. How is the coordination of motor control under so many degrees of freedom?
Response 4: We fully understand the reviewer’s concern and thank for this comment. We have added detailed information about the control platform configuration in Figure 3(b) and supplemented corresponding textual explanations in Figure 3(b) to elaborate on motor coordination mechanisms.
|
(b) |
Figure 3. (b) Configuration of the prosthetic hand control platform. The eight-channel sEMG signals collected by myo is directly sent to the PC, which is mapped to the expected force by the improved TCN trained in the PC. Meanwhile, the force signal collected by the thin-film pressure sensor and the current displacement of the electric cylinder are collected by STM32 and sent to the PC through the wireless module. After processing the actual force, actual displacement and expected force, the PC obtains an expected displacement. It is sent to the STM32 through the wireless module, and the STM32 drives the electric cylinder to move.
Comment 5: Is the TCN algorithm used for real-time processing of EMG signals from 8 channels? Suggest adding necessary explanations in the text on how the algorithm is specifically implemented.
Response 5: We fully understand the reviewer’s concern and thank for this comment. We have supplemented the text, which specifically includes:
(1) Supplement the text on line 222 “In this study, the improved TCN algorithm will be used to process the electromyographic signals from eight channels in real time, mapping the electromyographic signals to the expected grasping force of the prosthetic hand.” It is stated that the TCN algorithm will be used to process the electromyographic signals from 8 channels in real time.
(2) Supplement the experimental details in line 408, “Low-pass filtering is adopted to preprocess the original 8-channel sEMG signals to eliminate noise.”
(3) Supplement the experimental details on line 432, “These predicted forces served as real-time input commands for the control system.”
These supplementary details, in combination with the introduction in Section 2.4, should be able to clarify how the TCN algorithm processes 8-channel electromyographic signals in real time and its role in the system, providing a more complete technical background for understanding the prosthetic hand control system. If you need to elaborate further on any part of the implementation, please feel free to contact us at any time.
Comment 6: Is the signal from the IMU sensor used for closed-loop control of the movement of the Reiki hand when " The joint position data corresponding to different push 239 rod elongations are collected by an inertial measurement unit (IMU)," is mentioned in the article?
Response 6: We fully understand the reviewer’s concern and thank for this comment. We clarify that the signal from the IMU is not used for the closed-loop control of the prosthetic hand movement. Due to the compact structural design of the prosthetic hand, position sensors cannot be directly installed on the knuckles of the fingers, making the real-time position unknown. To solve this problem, we established a mathematical model (introduced in Section 2.5) to indirectly infer the joint rotation angle from the displacement of the electric cylinder push rod. The IMU is specifically used to verify the accuracy of this mathematical model. We installed the IMU on the knuckles of the prosthetic hand to collect the actual rotation angle of the joint during the displacement of the electric cylinder push rod. By comparing the joint rotation angle inferred by the mathematical model with the rotation angle data measured by the IMU, the rotation angle prediction error of the model was quantified, confirming that the mathematical model can reliably estimate the unmeasurable phalangeal positions.
To avoid ambiguity, we have supplemented this explanation in the corresponding part of the manuscript (in line 279 of Section 2.5), clarifying that IMU serves as a “tool for verifying model accuracy”.
Comment 7: According to the running results shown in Fig.8, after opening parameter k, the absolute error significantly increased. Please explain the detailed reasons in conjunction with the proposed control algorithm.
Response 7: We fully understand the reviewer’s concern and thank for this comment. We are very glad to have the opportunity to combine the logic of the control algorithm we proposed to explain the reasons for the significant increase in absolute error without k:
First of all, we need to clarify the core role of parameter k: k is a correction coefficient designed to compensate for the deviation between the ideal kinematic model of the prosthetic hand and its actual mechanical performance. Our PIP joint angle control algorithm was initially based on an ideal kinematic model - this model assumes that the angle of the PIP joint can be calculated through the elongation of the electric cylinder push rod. However, in practical applications, manufacturing errors (such as the length of the push rod and minor deviations in the processing of the joint shaft) and assembly errors (such as minor gaps at the joint connection and uneven friction distribution) can cause the actual PIP joint angle to deviate from the model's predicted value. The IMU sensor is used to collect the actual PIP joint angle during movement. The parameter k is derived from the ratio of the actual angle measured by the IMU to the angle predicted by the model, which is used to quantify and compensate for this mechanical deviation.We hope this explanation can clarify the logical connection among parameter k, control algorithm and error variation.
To enhance readability and avoid ambiguity, we have also increased the font size of the labels and annotations in Figure 8.

Reviewer 2 Report
Comments and Suggestions for Authors
The manuscript "An Adaptive Grasping Multi-Degree-of-Freedom Prosthetic Hand with A Rigid-Flexible Coupling Structure" shows construction of a robotic hand with multi degree of freedom with added control algorithm. In general this paper reads like a technical design work with main issue there are no scientific research implemented. The main drawback are that no discussion is provided either in result or in discussion section. Please provide a meaningful discussion to other works done in the field. This is a drawback of the general interesting new design of the robotic hand. Main parts:
- The Introduction needs a bit of state of the art literature in view of science how robotic hands of other researcher are designed and where are the flaws. Are those DOFs of the figures make the differences for grasping an object? Please give more details in such as well highlight your new design and what application such might fill (this is kind of missing).
- The Material part gives a detail overview how the robotic hand is build but under some figure such as Figure 2 (mostly all schemes at that stage, Figure 2 - Figure 6) there need to be given more information in the capture what those Figure shows. The one you used are too broad. Please correct such.
- Figure 8 and 9 can be merged together. Why has the absolute force such high fluctuations? The force is not covered over any theoretical value. Please explain this behavior of force.
- The only part that match for sensors are commercial sensors used. But in general for the DOFs robotic hand what is the sensibility of touch and the extent grasping larger objects? Also from Figure 13 -15 the actual force measured and predicted values are so different. Why does such happen? Please provide some kind of discussion of it (with references of other work who faced same issue).
- From the data provided in Table 1 there are no standard deviation shown. In presenting scientific or technical work the reproducibility of the result should be demonstrated in a meaningful statistic. Please provide such for you own work.
- The discussion section does not involve any comparison of other work made in the field. It would be helpful adding a new Table of comparison of other robotic hand designed research work showing the DOFs parameter, highlighting also your own achievements.
Author Response
Dear reviewer #2,
Thank you very much for giving us an opportunity to revise the manuscript sensors-3858169. According to your valuable comments, the manuscript has been carefully revised, and the major revisions were marked in red. We sincerely thank you for your careful read of the paper and helpful suggestions in improving the quality of our submission. The following are the detailed replies to your comments and the changes made in the revision:
Comment 1: The Introduction needs a bit of state of the art literature in view of science how robotic hands of other researcher are designed and where are the flaws. Are those DOFs of the figures make the differences for grasping an object? Please give more details in such as well highlight your new design and what application such might fill (this is kind of missing).
Response 1: Thank you for your constructive suggestions on enriching the introduction with the latest literature, freedom analysis and design application focus. Based on the suggestions, we have made extensive revisions to the Introduction to address all the aforementioned issues in detail and supplemented the following key points: The latest literature review on manipulator design and existing deficiencies: We systematically reviewed the representative studies on dexterous hands and prosthetic hands recently published in authoritative journals, and summarized their degrees of freedom, design features, contributions, and inherent limitations. At the same time, at the end of the Introduction, we clearly expounded on the core innovation of our prosthetic hand and the application scenarios it fills. The following is the entire content of the revised part:
For full-drive solutions, such as the Shadow Dexterous Hand (24 DOFs) have achieved human-like manipulation. However, their weight and volume are far greater than those of a natural human hand. Additionally, their complex tendon-driven mechanisms require frequent maintenance, making them difficult to be deployed as practical prosthetic hands. In contrast, the prosthetic hand(19 DOFs) developed by Hao Yang realizes muscle-like contraction through closed-loop temperature control, thereby resolving the weight-flexibility trade-off inherent in traditional high-DOFs prosthetic hands[3]. However, its actuators remain concentrated in the forearm and fail to achieve modularity.
Regarding underactuated solutions, devices like the QB SoftHand (19 DOFs) simplify the mechanics by using a single motor to drive all fingers. Nevertheless, they fail in tasks that demand selective finger control, as their rigid transmissions cannot adapt to irregular object contours. Another design, SoftHand-A (2 DOFs), provides a solution for developing low-cost, minimally actuated hands but is only capable of grasping lightweight objects [4].
Rigid-flexible coupled prosthetic limbs typically achieve human-like grasping functions while maintaining a size similar to that of a natural human hand. Examples include the prosthetic hand (5 DOFs) developed by Xuan et al., whose finger joints adopt a rigid skeleton integrated with a flexible driving structure, enabling high adaptive grasping capability [5]. The OLYMPIC Hand (5 DOFs) addresses the design issues and limitations of current modular commercial and research-grade prosthetic hands, exhibiting excellent performance in grasping [6]. However, both designs suffer from a low number of DOFs and face challenges in achieving precise force control.
Currently, commercial and research-grade prosthetic hands rely heavily on sEMG signals. However, these systems exhibit high variability due to muscle fatigue, changes in skin-electrode contact, and environmental interference. Research by Eric et al. investigated the impact of feedback mechanisms on the control accuracy of myoelectric prostheses, noting that traditional myoelectric control systems suffer from error accumulation issues in virtual tasks [7]. Simon et al. mapped EMG signals to force signals using a three-dimension TCN, leveraging the TCN's advantage of longer memory capacity compared to other recursive structures. This approach achieved a percentage root mean square error (%RMSE) of 29.2±13.1% and a coefficient of determination (R²) of 69.4±25.7% [8].
For researchers, grasping flexible objects with varying stiffness represents an important yet challenging task, as prosthetic hands struggle to autonomously determine the required degree of finger flexion to achieve the expected grasping force. Pimpalkar et al. noted in their research that the human hand can determine object stiffness within 15 milliseconds using vibrational signals at the moment of contact [9]. In contrast, existing prosthetic hands lack real-time stiffness sensing capabilities, resulting in delayed adjustment of grasping force relative to the object's deformation speed. Traditional methods primarily rely on tactile force sensors or measuring hand displacement under applied force, but these approaches only become effective after full contact is established [10], increasing the risk of damaging fragile objects during delicate tasks.
To address this series of challenges, we have made the following innovations:
- this study drew inspiration from the structures of rigid dexterous hands and rope-driven dexterous hands [4], and combined the advantages of rigid dexterous hands and rope-driven dexterous hands to design a rigid-flexible coupling structure. This structure makes the joints of the fingers of the dexterous hand consistent with those of the human hand, and the degrees of freedom of the metacarpophalangeal (MCP) joints are directly driven by miniature electric cylinders. The proximal interphalangeal (PIP) joint is indirectly driven by a planar four-bar mechanism, and the Angle corresponds to that of the MCP joint. The distal interphalangeal joint is driven by a rigid-flexible coupling structure, which can adapt to the shape of the object without active control, thereby increasing the grasping area and enhancing the grasping stability.
- Meanwhile, a TCN-LIA-MPC control algorithm is proposed. This algorithm enables the sEMG from the human forearm to be mapped into the human-expected grasping force via the Improved TCN algorithm. Subsequently, it obtains the expected displacement of the electric cylinders for grasping flexible objects using the linear iterative approximator (LIA) algorithm. Finally, it achieves effective tracking of the expected force through the Model Predictive Control (MPC) algorithm, thereby assisting humans in better grasping flexible objects.
.
Comment 2: The Material part gives a detail overview how the robotic hand is build but under some figure such as Figure 2 (mostly all schemes at that stage, Figure 2 - Figure 6) there need to be given more information in the captions what those Figure shows. The one you used are too broad. Please correct such.
Response 2: We fully understand the reviewer’s concern and thank for this comment. We fully admit that the original titles of Figures 2 to 6 were too broad. To address this issue, we have comprehensively revised these titles and the content of the images to convey the complete technical background without overly relying on the main text. The following are the corrections from Figures 2 to 6, which reflect the main improvements:
Modification description of Figure 2: Figure 2 has added an introduction to the position, number and function of the actuators and sensors in the picture
Figure 2. Schematic diagram of the overall structure of the prosthetic hand. The prosthetic hand has five fingers. Among them, the thumb is driven by the electric cylinder LAS10-023D, and the relative rotation of the palm and thumb is driven by LAS16-023D. The remaining four fingers are driven by the electric cylinder LA16-023D and the motor CHF-GW12T-10BY. Thin-film pressure sensors are evenly distributed on the three knuckles of the five fingers.
Modification note for Figure 3: Supplement the configuration of the dexterous hand control platform. And introduce how the coordination of motion control is carried out.
Figure 3. (a)The prototype of prosthetic hand. (b) Configuration of the prosthetic hand control platform. The eight-channel sEMG signals collected by myo is directly sent to the PC, which is mapped to the expected force by the improved TCN trained in the PC. Meanwhile, the force signal collected by the thin-film pressure sensor and the current displacement of the electric cylinder are collected by STM32 and sent to the PC through the wireless module. After processing the actual force, actual displacement and expected force, the PC obtains an expected displacement. It is sent to the STM32 through the wireless module, and the STM32 drives the electric cylinder to move.
Modification description of Figure 4: Clear labels have been added to the key components in Figure 4. Including clearly marking the positions of metacarpophalangeal joint (MCP), Proximal interphalangeal joint (PIP), and Distal Interphalangeal joint (DIP). Avoid blurring the structural details of the robot hand to ensure readability while enhancing the information value of the graphics. At the same time, the routing of the resistant tendon and the driving tendon was introduced.
Figure 4. Schematic diagram of the rigid-flexible coupling finger structure. The antagonistic tendon is located above,it starts from Shell 0, successively bypasses Optical Axis 1, Optical Axis 2, Optical Axis 3, and Optical Axis 4, and reaches Shell 7. It is responsible for resetting the fingers. Below is the driving tendon, which starts from Shell 0, passes around Optical Axis 1, Optical Axis 5, Optical Axis 6, and Optical Axis 4 in sequence, and reaches Shell 7. A spring is connected between the Optical Axis 6 and the Optical Axis 4. It is responsible for achieving the inward rotation of the end joint.
Modification notes for Figure 5: Figure 5 has added specific descriptions of input and output, making the expression no longer broad.
Figure 5. (a)The overall architecture of improved TCN. Input eight channels of sEMG signals and output the expected force. (B) Temporal convolutional block structure
Modification description of Figure 6: Figure 6 has added explanations for the colors and objects represented by the figures in the figure.
Figure 6. Simplified diagram of mechanical finger grasping force. Among them, the blue part represents the problem of being grasped, and the orange part represents the palm and fingers of the prosthetic hand.
Comment 3: Figure 8 and 9 can be merged together. Why has the absolute force such high fluctuations? The force is not covered over any theoretical value. Please explain this behavior of force.
Response 3: We fully understand the reviewer’s concern and thank for this comment.
First, regarding figure merging: We have combined Figure 8 and Figure 9 into a single figure (now labeled “Figure 8”)
Second, we are very sorry for the misunderstanding caused by the small text in our picture. Figure 8 is used to show the absolute value error between the theoretical angle and the actual angle of the PIP joint and the DIP joint. I would like to explain to you here why the absolute angle fluctuates so highly:First of all, we need to clarify the core role of parameter k: k is a correction coefficient designed to compensate for the deviation between the ideal kinematic model of the prosthetic hand and its actual mechanical performance. Our PIP joint angle control algorithm was initially based on an ideal kinematic model - this model assumes that the angle of the PIP joint can be calculated through the elongation of the electric cylinder push rod. However, in practical applications, manufacturing errors (such as the length of the push rod and minor deviations in the processing of the joint shaft) and assembly errors (such as minor gaps at the joint connection and uneven friction distribution) can cause the actual PIP joint angle to deviate from the model's predicted value. Similarly, DIP joints are also affected by these factors. The IMU sensor is used to collect the actual PIP joint angle during movement. The parameter k is derived from the ratio of the actual angle measured by the IMU to the angle predicted by the model, which is used to quantify and compensate for this mechanical deviation.
To enhance readability and avoid ambiguity, we have also increased the font size of the labels and annotations in Figure 8.
Comment 4: The only part that match for sensors are commercial sensors used. But in general for the DOFs robotic hand what is the sensibility of touch and the extent grasping larger objects? Also from Figure 13 -15 the actual force measured and predicted values are so different. Why does such happen? Please provide some kind of discussion of it (with references of other work who faced same issue).
Response 4: We fully understand the reviewer’s concern and thank for this comment. We have supplemented the relevant details. The sensibility of touch of the mechanical hand depends on the thin-film pressure sensor, which ranges from 5g to 2kg. After testing, it was found that the prosthetic hand could grasp a cylinder with a maximum diameter of approximately 11cm. This information will be supplemented in Section 2.2 (in line 195 and 201).
Meanwhile, in the discussion section, we made supplements, combining the research of other researchers to analyze why the actual measured force and the predicted value are quite different, as follows:
Despite the significant improvement in control accuracy, persistent fluctuations in the force signal were observed during experiments. We attribute this phenomenon to two primary mechanisms:
(1) Lack of multi-finger coordination: During grasping, independent control of the five fingers lacks a force coordination algorithm. When the deformable object shifts within the hand, asynchronous changes in contact conditions across the fingertip film pressure sensors lead to conflicts in force distribution;
(2) The sensitivity of the sensors used is relatively low; consequently, the minimum measurable contact force may already cause significant displacement of the object. Therefore, the actually measured force may be inaccurate, and errors may further be introduced due to the low sensor sensitivity [20].
(3) In traditional force-position control, unknown or varying environmental stiffness leads to steady-state force errors [21].
In future work, we will incorporate methods proposed by other researchers to address this issue. For instance, we will adopt their proposed unscented Bayesian optimization method to systematically tackle the uncertainties introduced by sensors; meanwhile, we will utilize a grey prediction model to predict environmental stiffness in real time, thereby reducing force tracking errors.
Comment 5: From the data provided in Table 1 there are no standard deviation shown. In presenting scientific or technical work the reproducibility of the result should be demonstrated in a meaningful statistic. Please provide such for you own work.
Response 5: We fully understand the reviewer’s concern and thank for this comment. We fully agree that providing standard deviation is crucial for demonstrating the reliability and reproducibility of scientific results, which is a core requirement of the technical work of mechanical hand performance evaluation.
As required, we have now supplemented the standard deviation data in the "Standard Deviation" dedicated column of Table 1(in line 467).
These standard deviation values clearly quantify the consistency of our experimental results and provide a clear reference for readers to evaluate the repeatability of the proposed prosthetic hand performance. I also discussed this data information in the final discussion(in line 473 and 481).
Comment6: The discussion section does not involve any comparison of other work made in the field. It would be helpful adding a new Table of comparison of other prosthetic hand designed research work showing the DOFs parameter, highlighting also your own achievements.
Response 6: We fully understand the reviewer’s concern and thank for this comment. This helps contextualize our prosthetic hand’s innovations and highlight its position in the field. As recommended, we have added Table 2 “Comparison of high-DOFs bionic hands.” in Section of the Discussion, which systematically contrasts our design with 6 representative research-based or commercial prosthetic hands or dexterous hands, focusing on DOFs and the number of actuators.
Table 2. Comparison of high-DOFs bionic hands.
|
Name |
Type |
Author/Company |
Number of DOFs |
Number of Actuators |
|
Our prosthetic hand |
Rigid-flexible coupled prosthetic hand |
Wu Longhan et al |
14 |
10 |
|
Xuan’s prosthetic hand |
Rigid-flexible coupled prosthetic hand |
Sicheng Xuan et al |
5 |
5 |
|
OLYMPIC Hand |
Rigid-flexible coupled prosthetic hand |
Liow Lois et al |
5 |
5 |
|
Schunk SVH |
Rigid dexterous hand |
SCHUNK |
9 |
9 |
|
Shadow hand |
Tendons drive dexterous hands |
Shadow Robot |
24 |
20 |
|
SoftHand-A hand |
Underactuated dexterous hand |
Haoran Li et al |
5 |
2 |
|
QB SoftHand |
Underactuated dexterous hand |
qb robotics |
19 |
2 |
In terms of structure, Table 2 lists other research-based or commercial prosthetic hands or dexterous hands. Compared with some other rigid-flexible coupled prosthetic hands (e.g., Xuan’s prosthetic hand, OLYMPIC Hand), the prosthetic hand developed in this study exhibits more DOFs, theoretically enabling more types of movements, which will be explored in subsequent studies. In other fields, when compared with humanoid dexterous hands, although our prosthetic hand is inferior to those of some dexterous hands (e.g., Shadow Hand), it has advantages in terms of size and weight, and does not require driving a large number of actuators. This facilitates the modular design of the prosthetic hand and the development of control algorithms, making it more suitable for use as a human-wearable prosthetic limb. Meanwhile, the distal knuckles of the prosthetic hand in this study possess adaptive capabilities, overcoming the limitation that traditional rigid dexterous hands (e.g., Schunk SVH) are unable to adaptively grasp objects.

Reviewer 3 Report
Comments and Suggestions for Authors
My comments to enhance the structure of the manuscript are:
1) Although the authors itemized their contributions in the “Conclusions” section, listing them with bullet points at the end of the “Introduction” section would also be a good addition. The main point is to indicate them “explicitly.” In addition, please criticize the state-of-the-art more deeply so that your contributions can be well-understood by the readers.
- In the “Abstract” section, please define what TCN stands for. In fact, the authors do not even need to define an abbreviation for the pertinent term in the Abstract section, as they do not use it at least twice.
- The authors use two abbreviations (DOF and DoF) for the same term (degree of freedom). Please be consistent, select one, and use it throughout the manuscript. Besides, pay attention to using the abbreviations you have defined instead of their open forms. For example, the authors often use the term “degree of freedom,” even though they already have an abbreviated version of it.
- At the end of the “Introduction” section, the authors indicate that they proposed a TCN-LIN-MPC control algorithm, and list this as a contribution in the Conclusions section. Still, there is no evident discussion about it in the Introduction section. The readers do not even know what TCN-LIN-MPC stands for. Although the authors have a separate “Abbreviation” section at the end, they should still define abbreviations within the text, so the readers do not need to go back and forth to understand.
- A general comment is that please pay attention to the usage of abbreviations. In some places, the authors use an abbreviation, but the readers understand its meaning one or two pages later.
2) Wherever appropriate, please put more effort into emphasizing the innovative sides of the study. This comment is not about “summarizing” them at the end of the “Introduction” and “Conclusions” sections. Provide your readers with some hints throughout the manuscript, so that they can actually conclude something when they read contributions as a whole in the “Conclusions” section.
3) Lines 140 to 151: What was the motivation behind selecting specifically (at least a kind of) the listed actuators (angle, stroke, maximum thrusts, etc.)? Using “references” can be an alternative to convince the reader.
- Please use, for example, Shell 0, Optical Axis 1, Shell 7 instead of shell 0, optical axis 1, shell 7.
4) Section 4.2: Please define what TCN and EMG stand for, where they are first mentioned in the manuscript. Please do not define an abbreviation for RNNs, as it is not used at least twice.
- How did the authors define the “hyperparameters” of the proposed TCN-based model? Based on experience or the available literature? Are they optimized values (31, 16, 7, 8, etc.)?
- “The obtained force f is…” Should it be “F” instead of “f?” “The force analysis is shown in Figure X.” Please update the figure number. The same applies to Line 413 (Figure X and Table X).
- Is multiplying the actual value by a coefficient (k, Line 258) standard in such applications? Please use references to support this action.
5) Section 3: Did the authors calculate the “predicted value” with the help of an optimized model in Figure 13? How did they ensure that their model worked correctly?
- As much as possible, please provide some information about participants (Line 363). How many, etc.?
- “…and the sampling rate was uniformly set to 100 Hz.” How did the authors designate this value?
- “The measured %RMSE is 14.66%, and the Validation R2 is 0.7003…” How can one evaluate if these values are low or high without comparing them to a baseline (standard) value? In other words, why are they satisfying values for validation?
- There is no clear interpretation of Figure 15. What conclusion should one draw from it?
- Table 1: Please use the minus sign instead of a hyphen while indicating negative values.
- Sections 3 and 4 can be expanded to provide a deeper assessment.
- Please remove the unnecessary parts: Line 478 and Lines 531 to 532.
- Some parts of the text have different font styles. Please check and correct it if necessary.
Author Response
Dear reviewer #3,
Thank you very much for giving us an opportunity to revise the manuscript sensors-3858169. According to your valuable comments, the manuscript has been carefully revised, and the major revisions were marked in red. We sincerely thank you for your careful read of the paper and helpful suggestions in improving the quality of our submission. The following are the detailed replies to your comments and the changes made in the revision:
Comment 1: Although the authors itemized their contributions in the “Conclusions” section, listing them with bullet points at the end of the “Introduction” section would also be a good addition. The main point is to indicate them “explicitly.” In addition, please criticize the state-of-the-art more deeply so that your contributions can be well-understood by the readers.
- In the “Abstract” section, please define what TCN stands for. In fact, the authors do not even need to define an abbreviation for the pertinent term in the Abstract section, as they do not use it at least twice.
- The authors use two abbreviations (DOF and DoF) for the same term (degree of freedom). Please be consistent, select one, and use it throughout the manuscript. Besides, pay attention to using the abbreviations you have defined instead of their open forms. For example, the authors often use the term “degree of freedom,” even though they already have an abbreviated version of it.
- At the end of the “Introduction” section, the authors indicate that they proposed a TCN-LIN-MPC control algorithm, and list this as a contribution in the Conclusions section. Still, there is no evident discussion about it in the Introduction section. The readers do not even know what TCN-LIN-MPC stands for. Although the authors have a separate “Abbreviation” section at the end, they should still define abbreviations within the text, so the readers do not need to go back and forth to understand.
- A general comment is that please pay attention to the usage of abbreviations. In some places, the authors use an abbreviation, but the readers understand its meaning one or two pages later.
Response 1: We fully understand the reviewer’s concern and thank for this comment. For each point, we have made systematic revisions to the manuscript. The specific adjustments are as follows:
(1) In the introduction, the innovation points are clearly listed and the latest technologies are discussed and criticized. At the same time, an explanation of the TCN-LIN-MPC control algorithm is added. The specific modified contents are placed at the end of this response.
(2) To eliminate the problems existing in abbreviations, we conducted a comprehensive review of the manuscript:
- For all abbreviations, we ensure that the first appearance is displayed as the full name (abbreviation).
- Delete the redundant abbreviations.
- Unified the form of abbreviations.
The following is the modified part in the introduction:
For full-drive solutions, such as the Shadow Dexterous Hand (24 DOFs) have achieved human-like manipulation. However, their weight and volume are far greater than those of a natural human hand. Additionally, their complex tendon-driven mechanisms require frequent maintenance, making them difficult to be deployed as practical prosthetic hands. In contrast, the prosthetic hand(19 DOFs) developed by Hao Yang realizes muscle-like contraction through closed-loop temperature control, thereby resolving the weight-flexibility trade-off inherent in traditional high-DOFs prosthetic hands[3]. However, its actuators remain concentrated in the forearm and fail to achieve modularity.
Regarding underactuated solutions, devices like the QB SoftHand (19 DOFs) simplify the mechanics by using a single motor to drive all fingers. Nevertheless, they fail in tasks that demand selective finger control, as their rigid transmissions cannot adapt to irregular object contours. Another design, SoftHand-A (2 DOFs), provides a solution for developing low-cost, minimally actuated hands but is only capable of grasping lightweight objects [4].
Rigid-flexible coupled prosthetic limbs typically achieve human-like grasping functions while maintaining a size similar to that of a natural human hand. Examples include the prosthetic hand (5 DOFs) developed by Xuan et al., whose finger joints adopt a rigid skeleton integrated with a flexible driving structure, enabling high adaptive grasping capability [5]. The OLYMPIC Hand (5 DOFs) addresses the design issues and limitations of current modular commercial and research-grade prosthetic hands, exhibiting excellent performance in grasping [6]. However, both designs suffer from a low number of DOFs and face challenges in achieving precise force control.
Currently, commercial and research-grade prosthetic hands rely heavily on sEMG signals. However, these systems exhibit high variability due to muscle fatigue, changes in skin-electrode contact, and environmental interference. Research by Eric et al. investigated the impact of feedback mechanisms on the control accuracy of myoelectric prostheses, noting that traditional myoelectric control systems suffer from error accumulation issues in virtual tasks [7]. Simon et al. mapped EMG signals to force signals using a three-dimension TCN, leveraging the TCN's advantage of longer memory capacity compared to other recursive structures. This approach achieved a percentage root mean square error (%RMSE) of 29.2±13.1% and a coefficient of determination (R²) of 69.4±25.7% [8].
For researchers, grasping flexible objects with varying stiffness represents an important yet challenging task, as prosthetic hands struggle to autonomously determine the required degree of finger flexion to achieve the expected grasping force. Pimpalkar et al. noted in their research that the human hand can determine object stiffness within 15 milliseconds using vibrational signals at the moment of contact [9]. In contrast, existing prosthetic hands lack real-time stiffness sensing capabilities, resulting in delayed adjustment of grasping force relative to the object's deformation speed. Traditional methods primarily rely on tactile force sensors or measuring hand displacement under applied force, but these approaches only become effective after full contact is established [10], increasing the risk of damaging fragile objects during delicate tasks.
To address this series of challenges, we have made the following innovations:
- this study drew inspiration from the structures of rigid dexterous hands and rope-driven dexterous hands [4], and combined the advantages of rigid dexterous hands and rope-driven dexterous hands to design a rigid-flexible coupling structure. This structure makes the joints of the fingers of the dexterous hand consistent with those of the human hand, and the degrees of freedom of the metacarpophalangeal (MCP) joints are directly driven by miniature electric cylinders. The proximal interphalangeal (PIP) joint is indirectly driven by a planar four-bar mechanism, and the Angle corresponds to that of the MCP joint. The distal interphalangeal joint is driven by a rigid-flexible coupling structure, which can adapt to the shape of the object without active control, thereby increasing the grasping area and enhancing the grasping stability.
- Meanwhile, a TCN-LIA-MPC control algorithm is proposed. This algorithm enables the sEMG from the human forearm to be mapped into the human-expected grasping force via the Improved TCN algorithm. Subsequently, it obtains the expected displacement of the electric cylinders for grasping flexible objects using the linear iterative approximator (LIA) algorithm. Finally, it achieves effective tracking of the expected force through the Model Predictive Control (MPC) algorithm, thereby assisting humans in better grasping flexible objects.
Comment 2: Wherever appropriate, please put more effort into emphasizing the innovative sides of the study. This comment is not about “summarizing” them at the end of the “Introduction” and “Conclusions” sections. Provide your readers with some hints throughout the manuscript, so that they can actually conclude something when they read contributions as a whole in the “Conclusions” section.
Response 2: Thank you for embedding profound suggestions on research innovation throughout the manuscript. This ensures that readers gradually recognize the value of our work as they read, rather than merely seeing contributions in the summary section. We revised many chapters to demonstrate innovation by linking it with the limitations of existing work, design logic and experimental results. The following are the core adjustments and examples to illustrate how to naturally emphasize innovation in each part:
(1) List innovations directly in the form of points in the introduction.
(2) In Section 2.3, it is mentioned the structural innovation. “Inspired by this physiological phenomenon, this study proposes a novel rigid–flexible coupled robotic finger.”
(3) In Section 2.4, it is mentioned to improve TCN to demonstrate the differences from traditional methods. “In this study, the improved TCN algorithm will be used to process the sEMG signals from eight channels in real time, mapping the electromyographic signals to the ex-pected grasping force of the prosthetic hand.”
(4) In Section 2.6, it is mentioned the innovative use of the algorithm. "To address this issue" we innovatively employed a position predictor in the control of rigid-flexible coupled fingers. The core of the position predictor lies in the online linear iterative approximation algorithm.”
(5) The comparison with the results of other studies is mentioned in Section 3.1. “Compared with other researchers (e.g., Simon et al., who achieved %RMSE is 29.2% and R² is 64.9 in their study) the improved TCN algorithm exhibits better performance in mapping eight-channel sEMG signals to expected forces.”
(6) In the discussion section, include comparisons with others' prosthetic or dexterous hands.
Comment 3: Lines 140 to 151: What was the motivation behind selecting specifically (at least a kind of) the listed actuators (angle, stroke, maximum thrusts, etc.)? Using “references” can be an alternative to convince the reader.
- Please use, for example, Shell 0, Optical Axis 1, Shell 7 instead of shell 0, optical axis 1, shell 7.
Response 3: We fully understand the reviewer’s concern and thank for this comment. We have made the following modifications:
(1) The motivation for transmission using worm gear and worm structure is declared. “The self-locking mechanism can ensure that fingers do not rotate in the event of power failure or when grasping objects that are too heavy.”
(2) The motor for using this type of micro electric cylinder was declared. “The selected electric cylinder has the advantages of small size and sufficient thrust, which is conducive to making our prosthetic structure more compact.”
(3) Using “Shell 0, Optical Axis 1, Shell 7” instead of “shell 0, optical axis 1, shell 7”. At the same time, these descriptions are placed in the titles of the picture to facilitate readers' reading.
Comment 4: Section 4.2: Please define what TCN and EMG stand for, where they are first mentioned in the manuscript. Please do not define an abbreviation for RNNs, as it is not used at least twice.
- How did the authors define the “hyperparameters” of the proposed TCN-based model? Based on experience or the available literature? Are they optimized values (31, 16, 7, 8, etc.)?
- “The obtained force f is…” Should it be “F” instead of “f?” “The force analysis is shown in Figure X.” Please update the figure number. The same applies to Line 413 (Figure X and Table X).
- Is multiplying the actual value by a coefficient (k, Line 258) standard in such applications? Please use references to support this action.
Response 4: We fully understand the reviewer’s concern and thank for this comment. For each point, we have made systematic revisions to the manuscript. The specific adjustments are as follows:
(1) According to the suggestions, when first mentioned, we define temporal convolutional network (TCN) and surface electromyographic (sEMG) and remove redundant RNN abbreviations. In addition, we further investigated other terms to ensure that there would be no issue of duplicate definitions.
(2) The hyperparameters (values: 31, 16, 7, 8, etc.) of the model based on TCN are selected through continuous experimental attempts, and the one with the best effect is ultimately chosen.
(3) The force symbol f has been modified to F, (Figure X and Table X) has been modified to (Table 1) in line 467. We conducted a cross-check of the entire manuscript to ensure there were no other oversights or omissions.
(4) Regarding the rationality of the correction factor k, we supplemented relevant literature in the manuscript. In the paper, the author proved the necessity of the correction factors (such as the joint friction coefficient and the inertia matrix term) in the model and derived the explicit relationship between the model parameters and the motion. Experimental verification shows that after introducing the correction coefficient, the prediction accuracy of the model has been improved.
Comment 5: Section 3: Did the authors calculate the “predicted value” with the help of an optimized model in Figure 13? How did they ensure that their model worked correctly?
- As much as possible, please provide some information about participants (Line 363). How many, etc.?
- “…and the sampling rate was uniformly set to 100 Hz.” How did the authors designate this value?
- “The measured %RMSE is 14.66%, and the Validation R2 is 0.7003…” How can one evaluate if these values are low or high without comparing them to a baseline (standard) value? In other words, why are they satisfying values for validation?
- There is no clear interpretation of Figure 15. What conclusion should one draw from it?
- Table 1: Please use the minus sign instead of a hyphen while indicating negative values.
- Sections 3 and 4 can be expanded to provide a deeper assessment.
- Please remove the unnecessary parts: Line 478 and Lines 531 to 532.
- Some parts of the text have different font styles. Please check and correct it if necessary.
Response 5: We fully understand the reviewer’s concern and thank for this comment. We have made targeted revisions to each point as follows:
(1) The "predicted value" (the expected force during the grasping process) in Figure 13 is calculated using our improved TCN model. These details are now added to Section 3.1 to clarify the optimization and validation logic of the model.
To ensure the normal operation of the model, the following two-step verification process needs to be implemented:
- Offline model optimization. The TCN model is trained on a large number of sEMG-force datasets, during which hyperparameters are continuously modified to minimize %RMSE and maximize R².
- Online performance verification. After the training, we selected five subjects (three men and two women, aged 25 to 35, weighing 55 to 75 kilograms, with no history of forearm muscle injury or neuromuscular disease) for testing to confirm the real-time effectiveness. The key is to check whether the predicted force trend is consistent with the actual force trend.
(2) Regarding the supplementary information of the participants, it is provided in the above responses as “3 males and 2 females, aged 25-35, weighing 55-75 kilograms, with no history of forearm muscle injury or neuromuscular disease.”
(3) Due to the limited performance of STM32, it shows performance deficiencies when collecting a large number of signals and processing various information. After testing, it performs well when the collected signal is 100HZ. Therefore, the signal collection frequency is set at 100hz.
(4) Regarding %RMSE and R², we supplemented comparisons with other researchers to demonstrate the contribution of improved TCN to this study.
Compared with other researchers (e.g., Simon et al.) who achieved %RMSE is 29.2% and R² is 64.9 in their study the improved TCN algorithm exhibits better performance in mapping eight-channel sEMG signals to expected forces.
(5) The hyphens in Table 1 have been changed to minus signs.
(6) In Section 3, we introduce standard deviation, and at the same time add a discussion on standard deviation in the discussion section. In Section 4, we introduce a comparative discussion with other prostheses and dexterous hands.
Section 3:
Table 1. Comparative experiment.
|
Grasping object |
Mode |
Expectation Force |
RMSE |
Max Error |
Mean Error |
Standard Deviation |
|
Sponge block |
LIA+MPC |
2 N |
0.4209 |
1.0489 |
0.2150 |
0.3147 |
|
changing |
0.8068 |
2.2061 |
0.4337 |
0.6955 |
||
|
AC |
2 N |
0.9767 |
1.6973 |
0.5359 |
0.7742 |
|
|
changing |
1.7685 |
3.6155 |
-0.9912 |
0.6063 |
||
|
LIA+PID |
2 N |
0.5103 |
1.0300 |
0.2495 |
0.4701 |
|
|
changing |
1.7726 |
4.3854 |
0.2797 |
1.5344 |
||
|
Cotton doll |
LIA+MPC |
2 N |
0.4500 |
1.3273 |
0.1816 |
0.2494 |
|
changing |
1.3537 |
3.9274 |
0.5565 |
0.7517 |
||
|
AC |
2 N |
0.8077 |
2.4817 |
0.6917 |
0.5512 |
|
|
changing |
1.5234 |
2.5414 |
1.0891 |
0.7839 |
||
|
LIA+PID |
2 N |
0.9131 |
1.5438 |
0.3532 |
0.8556 |
|
|
changing |
1.6440 |
4.5594 |
0.7784 |
1.2610 |
||
|
Silica gel column |
LIA+MPC |
3 N |
1.0317 |
2.0128 |
-0.2345 |
0.6727 |
|
changing |
1.2478 |
2.2787 |
-0.6884 |
1.0170 |
||
|
AC |
3 N |
1.9314 |
2.7228 |
1.1697 |
0.9033 |
|
|
changing |
2.6348 |
4.9876 |
0.7574 |
0.7078 |
||
|
LIA+PID |
3 N |
1.0837 |
2.2783 |
0.6289 |
1.0410 |
|
|
changing |
2.0797 |
4.3806 |
0.3062 |
1.4421 |
||
|
Empty plastic water bottle |
LIA+MPC |
3 N |
1.3085 |
3.8693 |
-0.0790 |
0.7569 |
|
changing |
1.1668 |
3.3395 |
0.6302 |
1.0801 |
||
|
AC |
3 N |
1.7759 |
2.7886 |
1.0125 |
0.5789 |
|
|
changing |
1.1655 |
3.1328 |
0.4134 |
1.1949 |
||
|
LIA+PID |
3 N |
1.2319 |
2.4915 |
0.6946 |
0.9461 |
|
|
changing |
1.8022 |
3.8478 |
0.9297 |
1.3424 |
Experimental results (Table 1) demonstrate that, when grasping the same deformable object, the force tracking accuracy of the LIA-MPC hybrid control strategy is significantly superior to that of conventional admittance control and the LIA-PID scheme, regardless of constant-force or dynamic variable-force grasping. Taking sponge grasping as an example, under constant expected force, RMSE of LIA-MPC is reduced by approximately 56.9% compared to admittance control and by approximately 17.5% compared to the LIA-PID scheme. Under a constant expected force, the standard deviation of the LIA-MPC is reduced by approximately 59.35% compared to the admittance control and by approximately 33.05% compared to the LIA-PID scheme. These results indicate that both LIA and MPC can achieve better force tracking accuracy when grasping deformable objects, thereby validating the effectiveness of the proposed algorithm in scenarios involving deformable object manipulation.
Notably, control performance exhibits significant differences depending on the stiffness of the deformable object. For low-stiffness objects (e.g., sponge block, cotton doll), the force tracking RMSE reaches 0.4209 and 0.4500 and the standard deviation reaches 0.3147 and 0.2494, respectively; whereas for high-stiffness objects (e.g., silica gel column, empty plastic water bottle), the RMSE increases to 1.0317 and 1.3085 and the standard deviation increases to 0.6727 and 0.7569, respectively.
Section 4:
Table 2. Comparison of high-DOFs bionic hands.
|
Name |
Type |
Author/Company |
Number of DOFs |
Number of Actuators |
|
Our prosthetic hand |
Rigid-flexible coupled prosthetic hand |
Wu Longhan et al |
14 |
10 |
|
Xuan’s prosthetic hand |
Rigid-flexible coupled prosthetic hand |
Sicheng Xuan et al |
5 |
5 |
|
OLYMPIC Hand |
Rigid-flexible coupled prosthetic hand |
Liow Lois et al |
5 |
5 |
|
Schunk SVH |
Rigid dexterous hand |
SCHUNK |
9 |
9 |
|
Shadow hand |
Tendons drive dexterous hands |
Shadow Robot |
24 |
20 |
|
SoftHand-A hand |
Underactuated dexterous hand |
Haoran Li et al |
5 |
2 |
|
QB SoftHand |
Underactuated dexterous hand |
qb robotics |
19 |
2 |
In terms of structure, Table 2 lists other research-based or commercial prosthetic hands or dexterous hands. Compared with some other rigid-flexible coupled prosthetic hands (e.g., Xuan’s prosthetic hand, OLYMPIC Hand), the prosthetic hand developed in this study exhibits more DOFs, theoretically enabling more types of movements, which will be explored in subsequent studies. In other fields, when compared with humanoid dexterous hands, although our prosthetic hand is inferior to those of some dexterous hands (e.g., Shadow Hand), it has advantages in terms of size and weight, and does not require driving a large number of actuators. This facilitates the modular design of the prosthetic hand and the development of control algorithms, making it more suitable for use as a human-wearable prosthetic limb. Meanwhile, the distal knuckles of the prosthetic hand in this study possess adaptive capabilities, overcoming the limitation that traditional rigid dexterous hands (e.g., Schunk SVH) are unable to adaptively grasp objects.
- Remove the unnecessary parts such as lines 478 and lines 531 to 532.
- Unify the font style of the manuscripts.

Round 2
Reviewer 2 Report
Comments and Suggestions for Authors
The authors revised the manuscript and gave good response. No further comments.